# SHARP MONOCULAR VIEW SYNTHESIS IN LESS THAN A SECOND

**Lars Mescheder**     **Wei Dong**     **Shiwei Li**     **Xuyang Bai**     **Marcel Santos**

**Peiyun Hu**     **Bruno Lecouat**     **Mingmin Zhen**     **Amaël Delaunoy**

**Tian Fang**     **Yanghai Tsin**     **Stephan R. Richter**     **Vladlen Koltun**

Apple

## ABSTRACT

We present SHARP[1], an approach to photorealistic view synthesis from a single image. Given a single photograph, SHARP regresses the parameters of a 3D Gaussian representation of the depicted scene. This is done in less than a second on a standard GPU via a single feedforward pass through a neural network. The 3D Gaussian representation produced by SHARP can then be rendered in real time, yielding high-resolution photorealistic images for nearby views. The representation is metric, with absolute scale, supporting metric camera movements. Experimental results demonstrate that SHARP delivers robust zero-shot generalization across datasets. It sets a new state of the art on multiple datasets, reducing LPIPS by 25–34% and DISTS by 21–43% versus the best prior model, while lowering the synthesis time by three orders of magnitude.

## 1 INTRODUCTION

Imagine revisiting a precious memory captured on camera. What if technology could lift the scene out of the image plane, recreating the three-dimensional world as it was then, putting you back in the scene? High-resolution low-latency AR/VR headsets can convincingly present spatial content. 3D representations can also be rendered on handheld displays. Can these surfaces be used to reconnect us with our memories in new ways?

Recent advances in neural rendering (Tewari et al., 2022) have demonstrated remarkable success in synthesizing photorealistic views, but many of the most impressive results leverage multiple input images and conduct time-consuming per-scene optimization. We are interested in view synthesis from a single photograph,

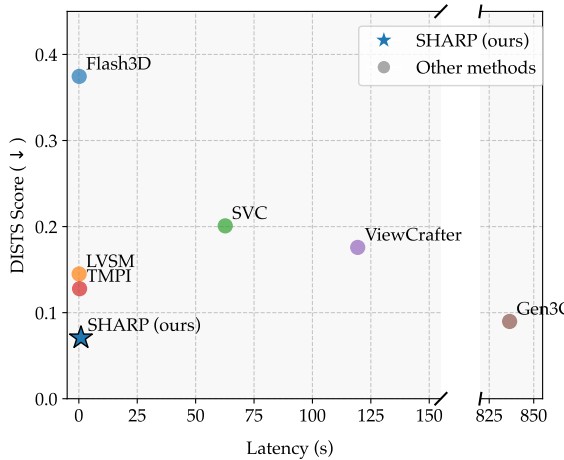

Figure 1: Synthesis time on a single GPU versus image fidelity on the ScanNet++ dataset.

to support real-time photorealistic rendering from nearby views. Specifically, our application setting yields the following desiderata. (a) Fast synthesis of a 3D representation from a single photograph, to support interactive browsing of personal photo collections. (b) Real-time photorealistic rendering

---

[1] https://github.com/apple/ml-sharp

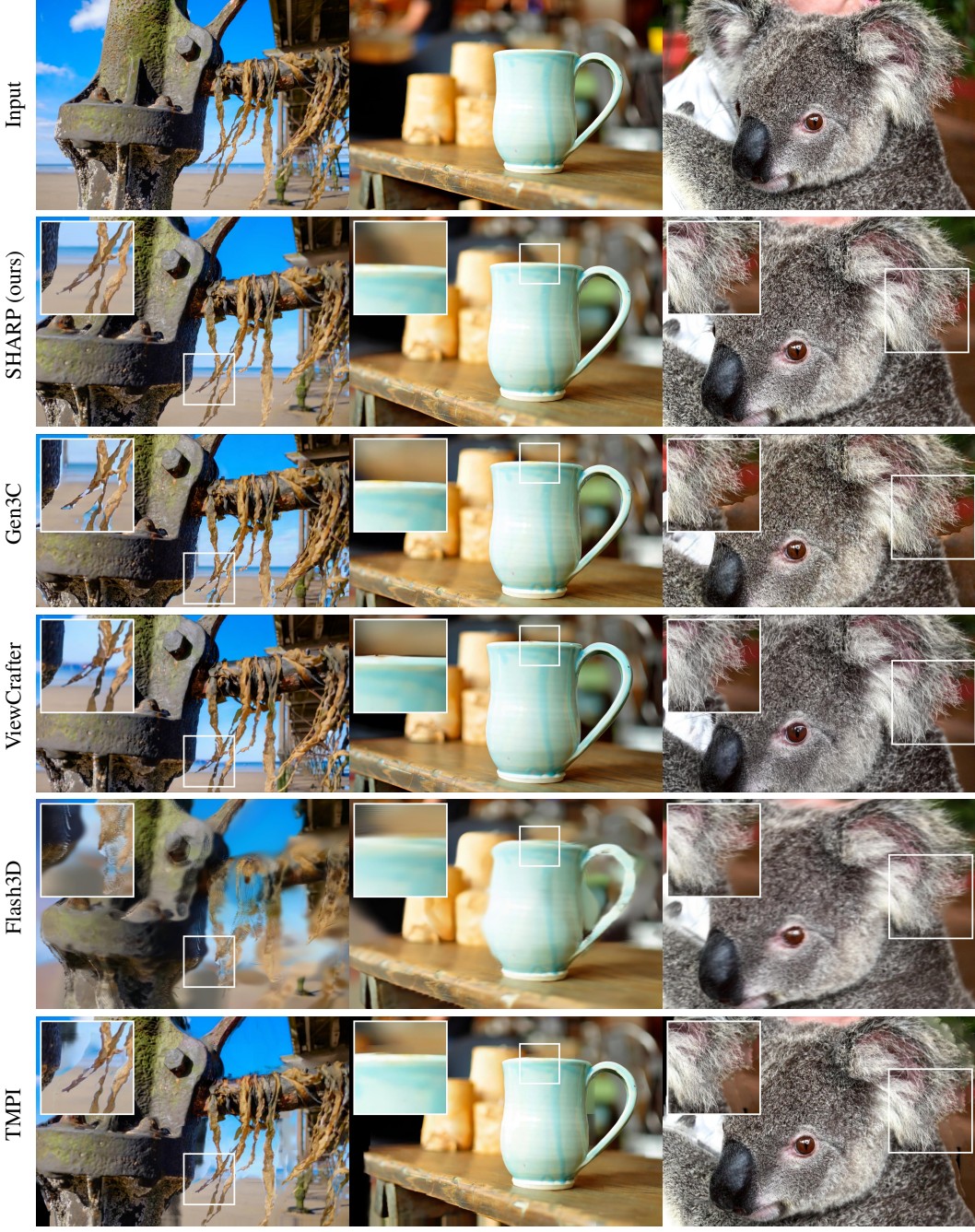

Figure 2: SHARP synthesizes a photorealistic 3D representation from a single photograph in less than a second. The synthesized representation supports high-resolution rendering of nearby views, with sharp details and fine structures, at more than 100 frames per second on a standard GPU. We illustrate on photographs from Unsplash (2022).

of the resulting 3D representation from nearby views. We wish to support natural posture shifts in AR/VR headsets, providing the experience of looking at a stable 3D scene from different perspectives, but need not support substantial travel ("walking around") within the photograph. (c) The 3D representation should be metric, with absolute scale, to accurately couple the virtual camera with a physical headset or another physical device.

In this paper, we present SHARP (Single-image High-Accuracy Real-time Parallax), our approach to meeting these desiderata. Given a photograph, SHARP produces a 3D Gaussian representation (Kerbl et al., 2023) of the depicted scene via a single forward pass through a neural network. This representation can then be rendered in real time from nearby views. Though the high-level approach (single image in, 3D Gaussian representation out) echoes prior work, SHARP delivers state-of-the-art visual fidelity while keeping the generation time under one second on an A100 GPU. (See Figure 1.) The key ingredients are scale and a number of technical choices whose importance we validate via controlled experiments.

First, we design a neural network that regresses a high-resolution 3D Gaussian representation from a single photograph. While our network comprises multiple modules, it is trained end-to-end to optimize view synthesis fidelity. Second, we introduce a carefully designed loss configuration that prioritizes the accuracy of synthesized views while regularizing away common artifacts. Third, we introduce a learned depth adjustment module that is used during training to facilitate view synthesis supervision in the presence of inaccurate depth estimates. Figure 2 shows some views synthesized by SHARP and a number of baselines.

We conduct a thorough experimental evaluation on multiple datasets that were not used during training, using powerful perceptual metrics such as LPIPS (Zhang et al., 2018) and DISTS (Ding et al., 2022) to assess image fidelity. SHARP improves image fidelity by substantial factors versus prior feedforward methods. In comparison to diffusion-based systems, SHARP delivers higher fidelity while reducing synthesis time by two to three orders of magnitude. Compared to the strongest prior method (Ren et al., 2025), SHARP reduces LPIPS by 25–34% and DISTS by 21–43% across the test datasets (in the zero-shot regime), while accelerating synthesis by three orders of magnitude and producing a 3D representation that supports high-resolution rendering of nearby views at 100 frames per second.

In summary, our contributions are as follows:

- **End-to-end architecture:** we design a novel network architecture that can be trained end-to-end to predict high-resolution 3D Gaussian representations.

- **Robust and effective loss configuration:** we carefully choose a series of loss functions to prioritize view synthesis quality while maintaining training stability and suppressing common visual artifacts.

- **Depth alignment module:** we introduce a simple module that can effectively resolve depth ambiguities during training, a fundamental challenge for regression-based view synthesis methods.

Using our insights, we demonstrate that state-of-the-art high-resolution view synthesis is feasible in a purely regression-based framework.

## 2 RELATED WORK

**View synthesis from multiple images.** Early image-based rendering approaches synthesized new views with minimal 3D modeling. Chen & Williams (1993) introduced view interpolation, enabling transitions between captured viewpoints. QuickTime VR (Chen, 1995) created navigable environments from panoramic images. Layered Depth Images (Shade et al., 1998) addressed occlusions by storing multiple depth values per pixel. Kang et al. (2006) survey the early years of image-based rendering.

More recently, deep learning and GPU acceleration transformed view synthesis from multiple images. Free View Synthesis (Riegler & Koltun, 2020) combined geometric scaffolds with learned features to synthesize novel views from distributed viewpoints. Stable View Synthesis (Riegler & Koltun, 2021) improved on this by enhancing stability and consistency across views. Neural radiance fields (NeRF) (Mildenhall et al., 2020) introduced continuous implicit representations that support remarkable levels of photorealism (Barron et al., 2023). 3D Gaussian Splatting (Kerbl et al., 2023) significantly accelerated rendering while maintaining visual fidelity through explicit 3D primitives. We use the 3D Gaussian representation developed by Kerbl et al. (2023), but apply it in the context of view synthesis from a single image.

A number of works develop feedforward prediction models for view synthesis from a small number of nearby views. IBRNet (Wang et al., 2021) generalized image-based rendering across scenes using learned features and ray transformers. MVSNeRF (Chen et al., 2021) reconstructed neural radiance fields from a few input images via cost volume processing. LaRa (Chen et al., 2024) regressed an object-level radiance field from sparse input images. GS-LRM (Zhang et al., 2024) leveraged a transformer to predict a 3D Gaussian representation from posed sparse images. Our work focuses on view synthesis from a single image.

**View synthesis from a single image.** The introduction of larger datasets (Zhou et al., 2018; Tung et al., 2024) enabled the transition from scene-specific multi-view optimization to learning-based pipelines that can infer a plausible 3D representation from a single image. Zhou et al. (2016) synthesized novel views from a single image through appearance flow. Subsequent work developed variants of depth-based warping (Wiles et al., 2020; Jampani et al., 2021) and multiplane images (MPI) (Zhou et al., 2018; Tucker & Snavely, 2020). AdaMPI (Han et al., 2022) adapted multiplane images to diverse scene layouts through plane depth adjustment and depth-aware color prediction, trained using a warp-back strategy on single-view image collections. Khan et al. (2023) proposed Tiled Multiplane Images (TMPI), which splits an MPI into many small tiled regions with fewer depth planes per tile, reducing computational overhead while maintaining quality. Several recent methods have drawn inspiration from the success of transformers in modeling long-range dependencies, leveraging large transformer-based encoder-decoder architectures to infer scene structure and appearance implicitly for novel views (Hong et al., 2024; Jin et al., 2025).

PixelNeRF (Yu et al., 2021) trained convolutional networks to predict an object-level neural radiance field from a single image. Splatter Image (Szymanowicz et al., 2024) introduced direct prediction of per-pixel Gaussians via a U-Net. Flash3D (Szymanowicz et al., 2025a) incorporated a pre-trained depth prediction network for generalization to more complex scenes. Schwarz et al. (2025) proposed a recipe for generating 3D worlds from a single image by decomposing this task into a number of steps and leveraging diffusion models.

Diffusion models have emerged as powerful tools for novel view synthesis with sparse input, offering high-quality results through iterative denoising processes (Po et al., 2024). Watson et al. (2023) developed an early application to view synthesis. Zero-1-to-3 (Liu et al., 2023) demonstrated zero-shot view synthesis by fine-tuning diffusion models on 3D object datasets. iNVS (Kant et al., 2023) repurposed diffusion inpainters for view synthesis by combining monocular depth estimation with inpainting. NerfDiff (Gu et al., 2023) distilled a 3D-aware diffusion model into NeRF by synthesizing virtual views to improve rendering under occlusion. These early attempts focused on object-level view synthesis.

At the scene level, ViewCrafter (Yu et al., 2025b), ZeroNVS (Sargent et al., 2024), CAT3D (Gao et al., 2024), SplatDiff (Zhang et al., 2025), Stable Virtual Camera (Zhou et al., 2025), Bolt3D (Szymanowicz et al., 2025b), Wonderland (Liang et al., 2025), WonderWorld (Yu et al., 2025a), See3D (Ma et al., 2025) and Gen3C (Ren et al., 2025) all applied diffusion models to view synthesis from sparse image sets or a single image. The diffusion-based approach supports impressive image quality from faraway viewpoints, leveraging diffusion priors to synthesize plausible appearance even for views that have no overlap with the input. On the other hand, image quality from nearby views (corresponding to natural head motion or posture shifts) can be noticeably less sharp and photorealistic than the input, while the synthesis time can sometimes stretch into minutes. (Although Bolt3D (Szymanowicz et al., 2025b) makes impressive progress on the latter front.) In contrast, we aim for real-time rendering of maximally photorealistic high-resolution images from nearby views, supporting a headbox that allows for natural posture shifts while maintaining photographic quality. Our approach generates a high-resolution 3D representation that provides such experiences from single-image input in less than a second on a single GPU, supporting conversion of pre-existing photographs to photorealistic 3D during interactive browsing of a photo collection.

## 3 METHOD

### 3.1 OVERVIEW

Our approach, SHARP, generates a 3D Gaussian representation from a single image via a forward pass through a neural network. The input to the network is a single monocular RGB image $\mathbf{I} \in$

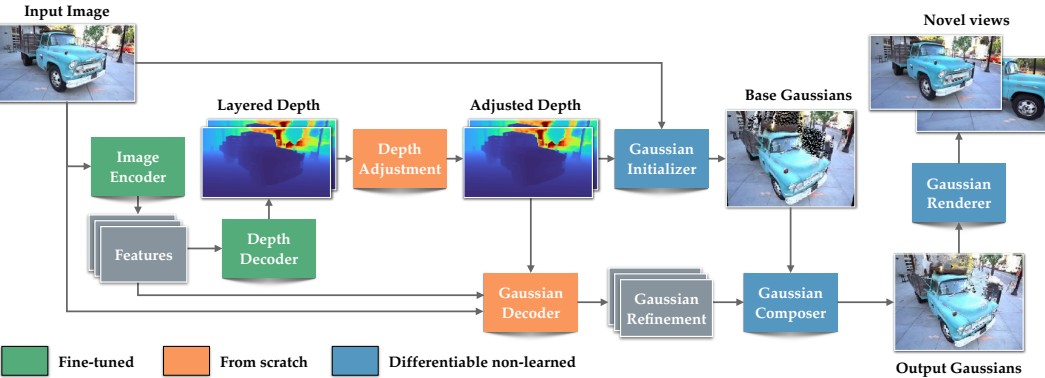

Figure 3: Our model consists of four learnable modules (Section 3.1): a pretrained encoder for feature extraction, a depth decoder that produces two distinct depth layers, a depth adjustment module, and a Gaussian decoder that refines all Gaussian attributes. The differentiable Gaussian initializer and composer assemble the Gaussians for the resulting 3D representation. The predicted Gaussians are rendered to the input and novel views for loss computation (Section 3.4).

$\mathbb{R}^{C \times H \times W}$, where $C = 3$ denotes the number of color channels, $H$ is the height, and $W$ is the width of the image. The output is a set of 3D Gaussians $\mathbf{G} \in \mathbb{R}^{K \times N}$, which can be rendered to arbitrary views using a differentiable renderer. Here $K = 14$ is the number of Gaussian attributes (3 for the position, 3 for scale, 4 for orientation, 3 for color and 1 for opacity) and $N$ is the number of output Gaussians. In practice, SHARP outputs $2 \times 768 \times 768 \approx 1.2$ million Gaussians per image, parameterized over a $768 \times 768$ grid with two layers. We do not use spherical harmonics (Kerbl et al., 2023), because the number of spherical harmonic coefficients grows quadratically with the order of the spherical harmonics and would lead to a large increase in output size. Figure 3 provides an overview of our method. The following paragraphs describe the modules in more detail.

**Monodepth backbone.** The input image $\mathbf{I} \in \mathbb{R}^{3 \times H \times W}$ is fed into a pretrained Depth Pro image encoder $\varphi_{\text{enc}}$ to produce 4 intermediate feature maps $(\mathbf{f}_i)_{i \in \{1,\dots,4\}} = \varphi_{\text{enc}}(\mathbf{I})$. As in Depth Pro (Bochkovskii et al., 2025), we resize the input image so that $H = W = 1536$. We then feed the intermediate feature maps into the Depth Pro decoder $\varphi_{\text{dec}}$ to produce a monocular depth map $\hat{\mathbf{D}} = \varphi_{\text{dec}}((\mathbf{f}_i)_{i \in \{1,\dots,4\}})$. Similar to Flynn et al. (2019), we duplicate the last convolutional layer of the decoder to produce a two-channel depth map $\hat{\mathbf{D}} \in \mathbb{R}^{2 \times H \times W}$.

One of our key observations is that depth is ill-defined and using a frozen monodepth model can degrade view synthesis fidelity, particularly for transparent or reflective surfaces (Wen et al., 2025). During training, we therefore unfreeze both $\varphi_{\text{dec}}$ and the low-resolution encoder part of $\varphi_{\text{enc}}$. This enables the full view synthesis training to adapt the depth prediction modules via backpropagation, in conjunction with downstream modules, for the end-to-end view synthesis objectives.

**Depth adjustment.** Although monocular depth estimation has made impressive advances in recent years, the depth estimator still needs to deal with the inherent ambiguity of the task. In monocular depth estimation, the network might just resolve the problem by predicting outputs at the mean scale of possible outcomes (Poggi et al., 2020). When depth estimates are used for view synthesis, however, this ambiguity can lead to visual artifacts.

To address this, we take inspiration from the line of work on Conditional Variational Autoencoders (C-VAE) (Sohn et al., 2015), which addresses the ambiguity by designing a posterior model. In a traditional C-VAE, the posterior would take ground-truth depth $\mathbf{D} \in \mathbb{R}^{H \times W}$ as input and produce a latent representation $\mathbf{z}$. During training this latent vector would be passed through an information bottleneck in the form of a KL divergence. This ensures that the latent represents the smallest amount of information required to resolve the ambiguity of the task. We simplify this scheme and adapt it to our setting by interpreting $\mathbf{z}$ as a scale map $\mathbf{S} \in \mathbb{R}^{H \times W}$ and replacing the KL divergence with a task-specific regularizer. More details are given in Section 3.4. The output of this module is an adjusted two-layer depth map $\bar{\mathbf{D}} = \mathbf{S}(\hat{\mathbf{D}}, \mathbf{D}) \odot \hat{\mathbf{D}}$.

**Gaussian initializer.** We use this adjusted two-layer depth map $\bar{\mathbf{D}} \in \mathbb{R}^{2 \times H \times W}$ and the input image $\mathbf{I} \in \mathbb{R}^{3 \times H \times W}$ to initialize a set of base Gaussians $\mathbf{G}_0 \in \mathbb{R}^{K \times 2 \times H' \times W'}$, where $H' = H/2$ and $W' = W/2$.

To compute $\mathbf{G}_0(\mathbf{I}, \bar{\mathbf{D}})$, we first subsample $\mathbf{I}$ and $\bar{\mathbf{D}}$ by a factor of 2, using average and min-pooling, respectively. This yields a downsampled depth map $\bar{\mathbf{D}}'$ and input image $\mathbf{I}'$. We then unproject the resulting depth map $\bar{\mathbf{D}}'$ to produce mean vectors $\mu(i,j) = [i \cdot \bar{\mathbf{D}}'(i,j), j \cdot \bar{\mathbf{D}}'(i,j), \bar{\mathbf{D}}'(i,j)]^T$. Note that we deliberately do not use the intrinsics matrix of the input image here. This enables the network to reason about Gaussian attributes in a normalized space without having to adapt its predictions to the field of view of the image. We set the scale proportional to depth: $s(i,j) = s_0 \cdot \bar{\mathbf{D}}'(i,j)$ with a fixed scale factor $s_0$. The color is initialized directly from the downsampled input image $c(i,j) = \mathbf{I}'(i,j)$. The rotation and opacity are initialized to a unit quaternion $[1,0,0,0]^T$ and a fixed value of $0.5$, respectively.

**Gaussian decoder.** While the initial Gaussians provide a reasonable starting point, they require substantial refinement to achieve high-fidelity rendering. The Gaussian decoder $\varphi_{\text{gauss}}$ takes as input the feature maps $(\mathbf{f}_i)_{i \in \{1,\dots,4\}}$ and the input image $\mathbf{I}$, and outputs refinements $\Delta \mathbf{G} \in \mathbb{R}^{K \times 2 \times H' \times W'}$ for all Gaussian attributes:

$$\Delta \mathbf{G} = \varphi_{\text{gauss}}((\mathbf{f}_i)_{i \in \{1,\dots,4\}}, \mathbf{I}). \tag{1}$$

These refinements include deltas for position $\Delta \mathbf{G}_{\text{pos}} \in \mathbb{R}^{3 \times 2 \times H' \times W'}$, scale $\Delta \mathbf{G}_{\text{scale}} \in \mathbb{R}^{3 \times 2 \times H' \times W'}$, rotation $\Delta \mathbf{G}_{\text{rot}} \in \mathbb{R}^{4 \times 2 \times H' \times W'}$, color $\Delta \mathbf{G}_{\text{color}} \in \mathbb{R}^{3 \times 2 \times H' \times W'}$, and opacity $\Delta \mathbf{G}_{\text{alpha}} \in \mathbb{R}^{1 \times 2 \times H' \times W'}$. The ability to refine Gaussians across all attributes is crucial for creating a coherent 3D representation that models detailed geometry and appearance.

**Gaussian composer.** The Gaussian composer takes the base Gaussians $\mathbf{G}_0 \in \mathbb{R}^{K \times 2 \times H' \times W'}$ and Gaussian refinements $\Delta \mathbf{G} \in \mathbb{R}^{K \times 2 \times H' \times W'}$ as input and produces the final Gaussian attributes $\mathbf{G} \in \mathbb{R}^{K \times 2 \times H' \times W'}$. Instead of directly adding the values, we compose them with an attribute-specific activation function $\gamma_{\text{attr}}$:

$$\mathbf{G}_{\text{attr}} = \gamma_{\text{attr}}\left( \gamma_{\text{attr}}^{-1}(\mathbf{G}_{0,\text{attr}}) + \eta_{\text{attr}} \Delta \mathbf{G}_{\text{attr}} \right). \tag{2}$$

The supplement provides the details on the activation functions $\gamma_{\text{attr}}$ and scale factors $\eta_{\text{attr}}$.

**Gaussian renderer.** The resulting Gaussian representation can be rendered from arbitrary viewpoints using an in-house differentiable renderer $\mathcal{R}$. The rendering process can be expressed as $\hat{\mathbf{I}} = \mathcal{R}(\mathbf{G}, \mathbf{P})$, where $\hat{\mathbf{I}}$ is the rendered image, $\mathbf{G}$ are the final Gaussian attributes, and $\mathbf{P}$ represents the camera projection parameters for the desired viewpoint. Since we predict the Gaussians in normalized space, we would theoretically need to transform them using the extrinsics and intrinsics of the source view. However, we can alternatively incorporate this transformation directly into the projection matrix for the target view: $\mathbf{P} = \mathbf{K}_{\text{tgt}} \mathbf{E}_{\text{tgt}} \mathbf{E}_{\text{src}}^{-1} \mathbf{K}_{\text{src}}^{-1}$, where $\mathbf{K}_{\text{src}}$ and $\mathbf{E}_{\text{src}}$ are the intrinsic and extrinsic matrices of the source view, and $\mathbf{K}_{\text{tgt}}$ and $\mathbf{E}_{\text{tgt}}$ are those of the target view. In contrast to image diffusion models, the inference cost is amortized: once a 3D representation is synthesized, it can be rendered in real time from new viewpoints.

## 3.2 NETWORK ARCHITECTURE

Our architecture includes a number of trainable modules, as illustrated in Figure 3. The complete network has approximately 340M trainable parameters (702M parameters in total). It processes a single $1536 \times 1536$ image and produces approximately 1.2 million Gaussians in under one second on a single GPU.

**Feature encoder.** We base our feature encoder on the Depth Pro backbone (Bochkovskii et al., 2025). This encoder processes the input image $\mathbf{I} \in \mathbb{R}^{C \times H \times W}$ and produces four feature maps $(\mathbf{f}_i)_{i \in \{1,\dots,4\}}$ at different resolutions. The Depth Pro backbone consists of two Vision Transformers (ViTs) (Dosovitskiy et al., 2021), one applied to a downscaled version of the input image and one applied to various image patches. The low-resolution image encoder and patch encoder each have 326M parameters. During training, we unfreeze the low-resolution image encoder to allow adaptation to the view synthesis task, while keeping the patch encoder and normalization layers frozen to preserve the pretrained feature extraction capabilities.

**Depth decoder.** Our depth decoder is based on the Dense Prediction Transformer (DPT) (Ranftl et al., 2021). We modify the original DPT decoder by duplicating the final convolutional layer to output two depth channels instead of one. Our decoder thus takes the feature maps $(\mathbf{f}_i)_{i \in \{1,\dots,4\}}$ from the encoder and produces a two-layer depth map $\hat{\mathbf{D}} \in \mathbb{R}^{2 \times H \times W}$. The first layer represents the primary visible surfaces, while the second layer may represent occluded regions and view-dependent effects. The depth decoder consists of multiple convolutional blocks with approximately 20M parameters. This module is fully unfrozen during training to optimize depth prediction for view synthesis.

**Gaussian decoder.** The Gaussian decoder predicts refinements for all Gaussian attributes. It has the same DPT architecture as the depth decoder but we replace the last upsampling block with a custom prediction head. The decoder takes as input the feature maps $(\mathbf{f}_i)_{i \in \{1,\dots,4\}}$, the input image $\mathbf{I}$, and the predicted depth maps $\hat{\mathbf{D}}$. It outputs a tensor $\Delta\mathbf{G} \in \mathbb{R}^{K \times 2 \times H' \times W'}$ that contains deltas for all Gaussian attributes: position (3 channels), scale (3 channels), rotation (4 channels), color (3 channels), and opacity (1 channel). This decoder has approximately 7.8M parameters and is trained from scratch. The high dimensionality of the output (approximately 16.5M values) enables fine-grained control over the Gaussian representation.

**Depth adjustment.** For the depth adjustment network we use a small U-Net (Ronneberger et al., 2015) with 2M parameters that takes both the predicted inverse depth $\hat{\mathbf{D}}^{-1}$ and the corresponding ground truth $\mathbf{D}^{-1}$ as inputs and produces a scale map $\mathbf{S} \in \mathbb{R}^{H \times W}$. During inference we replace the depth adjustment module with the identity function.

## 3.3 TRAINING STRATEGY

**Supervision.** We supervise predicted 3D Gaussians in image space through differentiable rendering. Each training sample consists of two views: the input view and the novel view. We predict Gaussians from the input view, render in both views, and evaluate losses on these renderings. The losses are defined in Section 3.4. We use a two-stage curriculum.

**Stage 1: Synthetic training.** We first train on synthetic data with perfect image and depth ground truth for both the input view and the novel view, allowing the network to learn fundamental principles of 3D reconstruction without real-world ambiguities. The synthetic data is further described in the supplement.

**Stage 2: Self-supervised finetuning (SSFT).** We fine-tune the model on real images that have no ground truth for view synthesis. To this end, we use our trained model to generate pseudo ground truth on single-view real images from OpenScene (2023) and online resources, detailed in the supplement. For each real image, we generate a 3D Gaussian representation and render a pseudo-novel view. We then use the pseudo-novel view as the input view, and the real input image as the novel view. The swapping of input and novel views forces the network to adapt to real images, enhancing its ability to generate coherent novel views.

Unlike AdaMPI (Han et al., 2022), which constructs stereo pairs from single-view collections using a warp-back strategy, our approach leverages the 3D representation generated by our model to create pseudo-novel views. This maintains geometric consistency while adapting to real images without requiring stereo pairs.

## 3.4 TRAINING OBJECTIVES

We train our network using a combination of loss functions:

**Rendering losses.** We apply an L1 loss between the rendered image $\hat{\mathbf{I}}$ and the ground truth $\mathbf{I}$ on both input and novel views:

$$\mathcal{L}_{\text{color}} = \sum_{\text{view} \in \{\text{input, novel}\}} \mathbb{E}_{p \sim \Omega}\left[|\hat{\mathbf{I}}_{\text{view}}(p) - \mathbf{I}_{\text{view}}(p)|\right], \tag{3}$$

where $\Omega$ denotes the set of all pixels $p$. We further use a perceptual loss (Johnson et al., 2016; Gatys et al., 2016; Suvorov et al., 2022) on novel views to encourage plausible inpainting:

$$\mathcal{L}_{\text{percep}} = \sum_{l=1}^{4} \lambda_l^{\text{feat}} \cdot \left\| \phi_l(\hat{\mathbf{I}}_{\text{novel}}) - \phi_l(\mathbf{I}_{\text{novel}}) \right\|^2 + \lambda_l^{\text{Gram}} \cdot \left\| M_l(\hat{\mathbf{I}}_{\text{novel}}) - M_l(\mathbf{I}_{\text{novel}}) \right\|^2, \quad (4)$$

where $\phi_l$ and $M_l$ are the $l$-th layer of our feature extractor and its Gram matrix, respectively. We apply a Binary Cross Entropy (BCE) loss to penalize rendered alpha on the input view to discourage spurious transparent pixels:

$$\mathcal{L}_{\text{alpha}} = \sum_{\text{view}\in\{\text{input, novel}\}} \mathbb{E}_{p\sim\Omega} \left[ \mathcal{L}_{\text{BCE}}(\hat{\mathbf{A}}_{\text{view}}(p), 1) \right], \quad (5)$$

where $\hat{\mathbf{A}}_{\text{view}}$ is the rendered alpha image.

**Depth losses.** We apply an L1 loss between the predicted and ground-truth disparity, only on the input view, exclusively on the first depth layer:

$$\mathcal{L}_{\text{depth}} = \mathbb{E}_{p\sim\Omega} \left[ |\bar{\mathbf{D}}_{(1)}^{-1}(p) - \mathbf{D}^{-1}(p)| \right], \quad (6)$$

where $\bar{\mathbf{D}}_{(1)}$ and $\mathbf{D}$ are the first predicted depth layer and the ground-truth depth, respectively.

**Regularizers.** We apply a total variation regularizer on the second depth layer to promote smoothness:

$$\mathcal{L}_{\text{tv}} = \mathbb{E}_{p\sim\Omega} \left[ |\nabla_x \bar{\mathbf{D}}_{(2)}^{-1}(p)| + |\nabla_y \bar{\mathbf{D}}_{(2)}^{-1}(p)| \right], \quad (7)$$

where $\bar{\mathbf{D}}_{(2)}$ is the second predicted depth layer. Additionally, we apply a regularizer to suppress floaters with large disparity gradients:

$$\mathcal{L}_{\text{grad}} = \mathbb{E}_{i\sim\mathcal{I}} \left[ \mathbf{G}_{\text{alpha}}(i) \cdot \left( 1 - \exp\left( -\frac{1}{\sigma} \max\left\{ 0, |\nabla\bar{\mathbf{D}}^{-1}(\pi(\mathbf{G}_0(i)))| - \epsilon \right\} \right) \right) \right], \quad (8)$$

where $\mathcal{I}$ is the index set for the Gaussians and $\pi(\cdot)$ computes the projection of the Gaussian position onto the 2D image plane. We use $\sigma = \epsilon = 10^{-2}$. We further constrain Gaussian offset magnitudes $\Delta\mathbf{G}_x, \Delta\mathbf{G}_y$ to discourage extreme deviations from the base Gaussians:

$$\mathcal{L}_{\text{delta}} = \mathbb{E}_{i\sim\mathcal{I}} \left[ \max\{|\Delta\mathbf{G}_x(i)| - \delta, 0\} + \max\{|\Delta\mathbf{G}_y(i)| - \delta, 0\} \right], \quad (9)$$

with $\delta = 400.0$. In screen space, we regularize the variance of projected Gaussians:

$$\mathcal{L}_{\text{splat}} = \mathbb{E}_{i\sim\mathcal{I}} \left[ \max\{\sigma(\mathbf{G}(i)) - \sigma_{\max}, 0\} + \max\{\sigma_{\min} - \sigma(\mathbf{G}(i)), 0\} \right], \quad (10)$$

where $\sigma(\cdot)$ computes the projected Gaussian variance and $\sigma_{\min} = 10^{-1}, \sigma_{\max} = 10^2$.

**Depth adjustment.** We regularize the depth adjustment with an MAE loss and a multiscale total variation regularizer:

$$\mathcal{L}_{\text{scale}} = \mathbb{E}_{p\sim\Omega} \left[ |\mathbf{S}(p) - 1| \right] \quad \text{and} \quad \mathcal{L}_{\nabla\text{scale}} = \sum_{k=1}^{6} \mathbb{E}_{p\sim\Omega_{\downarrow k}} \left[ |\nabla\mathbf{S}_{\downarrow k}(p)| \right]. \quad (11)$$

Here $\mathbf{S}_{\downarrow k}$ denotes a scale map downsampled by a factor $2^k$ on the downsampled image domain $\Omega_{\downarrow k}$. The depth adjustment losses act as an information bottleneck, encouraging the network to learn the most compact representation to resolve depth ambiguities.

The final loss is a composition of all the loss terms:

$$\mathcal{L} = \sum_{d\in\mathcal{D}} \lambda_d \mathcal{L}_d + \sum_{r\in\mathcal{R}} \lambda_r \mathcal{L}_r + \sum_{s\in\mathcal{S}} \lambda_s \mathcal{L}_s, \quad (12)$$

where $\mathcal{D} = \{\text{color, alpha, depth, percep}\}$, $\mathcal{R} = \{\text{tv, grad, delta, splat}\}$, $\mathcal{S} = \{\text{scale}, \nabla\text{scale}\}$ are the attribute sets for the data terms and regularizers. The hyperparameters are specified in the supplement.

## 4 EXPERIMENTS

We first train our model for 100K steps on 128 A100 GPUs using synthetic data only (Stage 1). We then fine-tune our model using self-supervision for 60K steps on 32 A100 GPUs (Stage 2).

**Datasets.** We evaluate our approach on multiple datasets with metric poses: Middlebury (Scharstein et al., 2014), Booster (Ramirez et al., 2024), ScanNet++ (Yeshwanth et al., 2023), WildRGBD (Xia et al., 2024), ETH3D (Schöps et al., 2017), and Tanks and Temples (Knapitsch et al., 2017). The sampling choices are discussed in the supplement. We do not include non-metric datasets such as RealEstate10K (Zhou et al., 2018).

**Evaluation metrics.** We employ LPIPS (Zhang et al., 2018) and DISTS (Ding et al., 2022) to quantitatively assess the quality of novel view synthesis. We focus primarily on these perceptual metrics, since older pointwise metrics such as PSNR and SSIM can be overly sensitive to small translations, where even a 1% shift can lead to catastrophic drops in scores despite visually similar results. (See the supplement for an illustration. We also list PSNR and SSIM numbers in the supplement for completeness.) Since we are interested in sharp high-resolution view synthesis, we evaluate all methods on the full-resolution ground truth. If a method generates results at lower resolution, we resize the output image to the input resolution before evaluation. If a method crops the input and generates cropped results (Ren et al., 2025; Yu et al., 2025b; Jin et al., 2025; Zhou et al., 2025), we evaluate against correspondingly cropped ground-truth images.

**Baselines.** We compare SHARP to the following state-of-the-art methods: Flash3D (Szymanowicz et al., 2025a), which is based on 3D Gaussians; TMPI (Khan et al., 2023), which uses multi-plane images; LVSM (Jin et al., 2025), which is based on image-to-image regression; and Stable Virtual Camera (SVC) (Zhou et al., 2025), ViewCrafter (Yu et al., 2025b), and Gen3C (Ren et al., 2025), which employ diffusion models.

**Quantitative evaluation.** Table 1 presents a quantitative evaluation of SHARP and the baselines in the zero-shot regime. (Cross-dataset generalization to datasets that were not used during training.) For each metric, we report the mean value over all test samples. SHARP achieves the highest accuracy on all metrics across all datasets. Additional experimental results are provided in the supplement.

Table 1: Quantitative evaluation. Lower is better. Best , second-best , and third-best in each column are highlighted.

| | Middlebury | | Booster | | ScanNet++ | | WildRGBD | | Tanks and Temples | | ETH3D | |
|---|---|---|---|---|---|---|---|---|---|---|---|---|
| | DISTS↓ | LPIPS↓ | DISTS↓ | LPIPS↓ | DISTS↓ | LPIPS↓ | DISTS↓ | LPIPS↓ | DISTS↓ | LPIPS↓ | DISTS↓ | LPIPS↓ |
| Flash3D | 0.359 | 0.581 | 0.409 | 0.370 | 0.374 | 0.572 | 0.159 | 0.345 | 0.382 | 0.683 | 0.535 | 0.651 |
| TMPI | 0.158 | 0.436 | 0.232 | 0.409 | 0.128 | 0.309 | 0.114 | 0.327 | 0.309 | 0.693 | 0.396 | 0.720 |
| LVSM | 0.274 | 0.555 | 0.307 | 0.404 | 0.145 | 0.302 | 0.095 | 0.257 | 0.227 | 0.575 | 0.555 | 0.664 |
| SVC | 0.208 | 0.629 | 0.283 | 0.448 | 0.201 | 0.596 | 0.157 | 0.531 | 0.230 | 0.733 | 0.420 | 0.708 |
| ViewCrafter | 0.373 | 0.751 | 0.318 | 0.523 | 0.176 | 0.526 | 0.148 | 0.386 | 0.295 | 0.759 | 0.454 | 0.748 |
| Gen3C | 0.164 | 0.545 | 0.207 | 0.384 | 0.090 | 0.227 | 0.106 | 0.285 | 0.177 | 0.566 | 0.408 | 0.734 |
| SHARP (ours) | 0.097 | 0.358 | 0.119 | 0.270 | 0.071 | 0.154 | 0.069 | 0.190 | 0.122 | 0.421 | 0.258 | 0.554 |

**Qualitative results.** Figure 2 shows novel views synthesized by SHARP and a number of baselines. Additional qualitative results, including on images from all evaluation datasets, can be found in the supplement. SHARP consistently produces higher-fidelity renderings from nearby views.

**Ablation studies.** We conduct extensive ablation studies and controlled experiments on the losses, training curriculum, depth adjustment, and more. The perceptual loss brings substantial improvement in visual quality, while the regularizers address some classes of artifacts. The learned depth adjustment boosts image sharpness and enhances details. The SSFT likewise yields crisper synthesized views. Detailed results and sample images are provided in the supplement.

## 5 CONCLUSION

We presented SHARP, an approach to real-time photorealistic rendering of nearby views from a single photograph. SHARP synthesizes a 3D Gaussian representation via a single forward pass through a neural network in less than a second on a standard GPU. This 3D representation can then be rendered in real time at high resolution from nearby views. Our experiments demonstrate

that SHARP delivers state-of-the-art image fidelity for nearby view synthesis, outperforming recent approaches that are in some cases two to three orders of magnitude more computationally intensive.

One clear opportunity for future work is to extend the methodology to support photorealistic synthesis of faraway views without compromising the fidelity of nearby views or the benefits of fast interactive synthesis. This may call for judicious integration of diffusion models (Po et al., 2024), possibly with the aid of distillation for reducing synthesis latency (Yin et al., 2024). With diffusion models, a unified view synthesis routine for single-view, multi-view, and video input (Ren et al., 2025; Ma et al., 2025) may emerge as a versatile generalization. Another interesting avenue is a principled treatment of view-dependent and volumetric effects (Verbin et al., 2024).

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

SUPPLEMENTARY MATERIAL

# A    IMPLEMENTATION DETAILS

## A.1    ATTRIBUTE SPECIFIC ACTIVATION

Activation functions $\gamma$ and their corresponding scale factors $\eta$ in Eq. 2 are specified below:

|  | position $(x/z, y/z)$ | position $(z^{-1})$ | color | rotation | scale | alpha |
|---|---|---|---|---|---|---|
| $\gamma$ | identity | softplus | sigmoid | identity | sigmoid | sigmoid |
| $\eta$ | $10^{-3}$ | $10^{-3}$ | $10^{-1}$ | 1 | 1 | 1 |

For the position, we apply the activation function in NDC space, *i.e.* we first map $[x, y, z] \rightarrow [x/z, y/z, 1/z]$ before applying the activation function and adding the delta. After the operation, we transform the result back to world coordinates.

## A.2    TRAINING OBJECTIVES

In the loss configuration, we choose $\lambda_{\text{color}} = 1.0, \lambda_{\text{alpha}} = 1.0, \lambda_{\text{percep}} = 3.0, \lambda_{\text{depth}} = 0.2, \lambda_{\text{tv}} = 1.0, \lambda_{\text{grad}} = 0.5, \lambda_{\text{delta}} = 1.0, \lambda_{\text{splat}} = 1.0, \lambda_{\text{scale}} = 0.1, \lambda_{\nabla \text{scale}} = 5.0$ in Eq. 12.

For the perceptual loss in Eq. 4 we use $\lambda_l^{\text{feat}} = \frac{1}{D_l \cdot H_l \cdot W_l}$ and $\lambda_l^{\text{Gram}} = \frac{10}{D_l^2}$, where $D_l \times H_l \times W_l$ denotes the shape of the $l$-th feature map $\phi_l(\cdot) \in \mathbb{R}^{D_l \times H_l \times W_l}$.

We trained the network using the Adam optimizer (Kingma & Ba, 2015) with a cosine learning rate schedule (Loshchilov & Hutter, 2017). The learning rate was linearly warmed up for 10,000 iterations to an initial value of $1.6 \times 10^{-4}$, after which it decayed to a final value of $1.6 \times 10^{-5}$.

## A.3    VIEW FRUSTUM MASKING

We implement a view frustum masking technique to address ambiguity in view synthesis, since regions occluded in the original view have multiple plausible reconstructions. By using depth information to determine which regions in the new view correspond to points visible in the original view, we apply supervision only where ground truth is reliable.

To calculate this mask, we project points from the target view back to the source view:

$$[x' \cdot z', y' \cdot z', z', 1]^T \xrightarrow{\mathbf{T}_{\text{novel} \rightarrow \text{source}}} [x \cdot z, y \cdot z, z, 1]^T. \tag{13}$$

The mask is then defined as

$$M(x', y') = \begin{cases} 1, & \text{if } -1.05 \leq x \leq 1.05 \text{ and } -1.05 \leq y \leq 1.05 \\ 0, & \text{otherwise} \end{cases} \tag{14}$$

This mask is applied to all image-based losses on the target view.

## A.4    THE PERCEPTUAL LOSS

Here we detail the challenges and our solutions in incorporating the perceptual loss. We employ the perceptual loss aimed at improving inpainting (Suvorov et al., 2022). Similar to its application in the image domain, we initially applied the loss only in the occluded image patches of the novel view; however, through experiments, we observed that applying the loss to the entire rendered image resulted in more plausible details and fewer artifacts in general, even in the non-occluded foreground regions, as seen in Figure 9.

However, this formulation of loss imposes two major challenges: (a) heavy memory overhead, and (b) compromised sharpness.

**Memory.** The perceptual loss maximizes feature similarity between the rendering and the ground truth. It is constructed from a combination of MSE losses on layer-wise feature maps of deep neural

networks (in our case, a ResNet-50). Since the loss itself is computed through a deep neural network, when applied to full images, it adds a significant memory overhead to the already large computation graph during backpropagation. Furthermore, when the loss is applied to both the reconstruction and synthesized views, the accumulated computation graph can lead to out-of-memory conditions even on an A100 with a generous memory pool (40GB) with a batch size of one.

To address the problem, one potential workaround would be simply reducing the activation precision to BF16; however, this does not address the fundamental problem of computation graph accumulation, prevents scaling the loss to more novel view supervisions, and causes training instability, especially when predicted 3DGS contains properties (*e.g.*, singular values) that are prone to precision changes. Gradient checkpointing is another option, but it can drastically impair training efficiency.

To address this problem, we propose a novel computation graph surgery mechanism. We implement a *surgery operator* to accept and cache gradients along with the inputs during the forward pass, and to inject cached gradients during the backward pass. Then, at the perceptual loss node in the graph, we *eagerly pre-compute* the gradients with respect to the features via an explicit *autograd* call, release the partial computation graph involving ResNet, and override the node with the surgery-operated one. This strategy avoids accumulating the computation graph and leads to a compact graph that is agnostic to the number of pixels or views. As a result, we are able to continue training at the full FP32 precision with perceptual loss on both reconstruction and novel views, without compromising training throughput. It is worth noting that the surgery operator is a general operator and can be integrated into any training framework with similar memory concerns regarding the computation graph.

**Sharpness.** Since the perceptual loss is applied to the latent feature space, while it offers the benefit of more plausible inpainting, the output renderings often tend to be blurry in the pixel space. Through backpropagation, this translates to large and blobby 3D Gaussians, whose renderings are simultaneously less detailed and more time-consuming.

To encourage sharpness, we explored losses that reduce feature space distance and revived the Gram matrix loss (Reda et al., 2022) that was originally designed for style transfer. This loss matches the auto-correlation of the latent features, further enhancing feature space similarity and boosting image sharpness. We introduce this loss in Eq. 4. As mentioned above, the original Gram matrix loss was applied to VGG features targeted at style transfer, and cannot be directly transferred to the ResNet-50 features pre-trained for inpainting. We conducted a series of controlled experiments with $\lambda_l^{\text{Gram}} = \frac{j}{D_l^2}, j \in \{1, 10, 100, 500\}$, along with $\lambda_l^{\text{feat}} = \frac{k}{D_l \cdot H_l \cdot W_l}, k \in \{0.1, 0.3, 1.0, 10.0\}$, and identified the most promising combination, as reported in Section A.2, through extensive quantitative metric validation and qualitative human inspection. This carefully tuned perceptual loss improves the DISTS metrics by 62% and 47% on benchmarks (as seen in Table 8), and reduces rendering latency by 49% and 36% respectively (Table 9).

# B TRAINING DATA

## B.1 SYNTHETIC DATA

In Stage 1 of training (Section 3.3), we use a large-scale synthetic dataset generated using an in-house procedural content generation system. This system operates by sampling from a large collection of artist-made environments, comprising over 2K outdoor and 5K indoor scenes, and augmenting them procedurally. For each sampled environment, the framework populates the scene with high-quality digital human characters featuring realistic hair grooms and garments, along with a variety of additional objects. This approach enhances the structural and visual diversity of the dataset while preserving the underlying artistic quality of the base environments.

To further enhance scene diversity and complexity, the framework supports random placement of various object types, including thin structures, transparent materials, and reflective surfaces, across a wide range of spatial configurations. It also offers fine-grained control over camera parameters such as position, orientation, and focal length, as well as detailed illumination settings. Lighting setups include physically-based direct light sources, with variations in direction, intensity, and color temperature. We also use high-dynamic-range (HDR) environment maps, which are sampled from a

curated collection of high-resolution HDRIs. This combination enables realistic global illumination effects under diverse and physically plausible lighting conditions.

For each scene, we identify one object of interest and position a ring containing 10 virtual cameras around it at varying distances and angles. The cameras are arranged in concentric circles such that the cameras are no more than 60 cm apart, simulating a multi-view capture setup. This allows the dataset to capture the same object or scene element from diverse perspectives. All images are rendered using the V-Ray physically based rendering engine, ensuring photorealistic lighting and material interactions. The final dataset consists of approximately 700K unique rendered scene instances, each with 11 rendered views, totaling around 8M images at $1536 \times 1536$ or $2048 \times 2048$ resolutions.

### B.2 REAL-WORLD DATA

In Stage 2 of training (Section 3.3), we use OpenScene (2023) as well as a collection of high-quality photographs from Shutterstock, Getty Images, and Flickr, all with commercial licenses. The dataset contains 2.65M images in total.

## C EXPLANATORY FIGURES

### C.1 IMAGE FIDELITY METRICS

To determine which metrics are most suitable for evaluating view synthesis quality, we conducted an experiment analyzing how different metrics respond to simple image translations.

As shown in Table 2 and Figure 4, we observe that older pointwise metrics such as PSNR and SSIM are highly sensitive to small spatial misalignments. A mere 1% translation causes PSNR to drop to 11.2 and SSIM to 0.375, values that are surprisingly close to those obtained when comparing with a mean image (PSNR 10.7, SSIM 0.351).

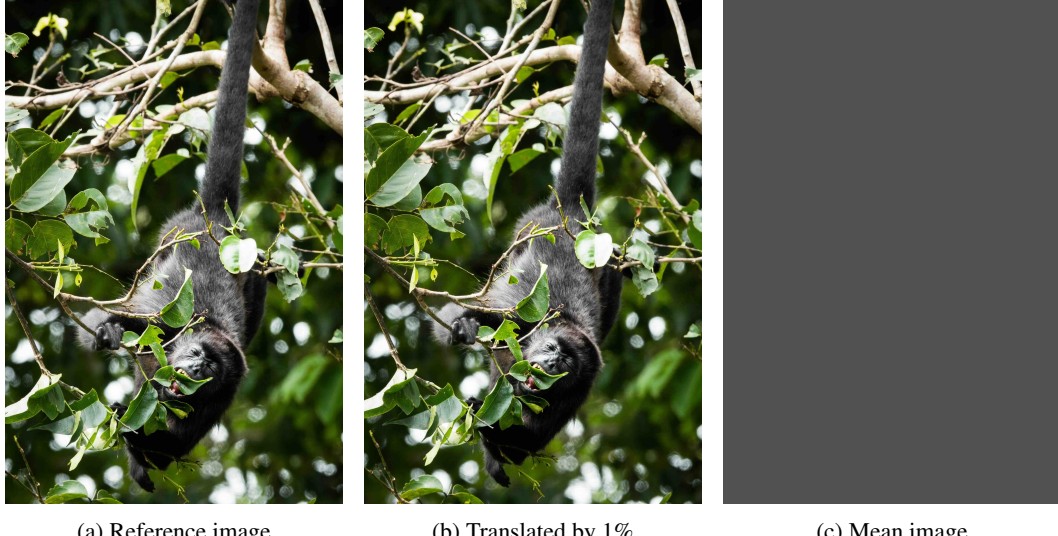

|           (a) Reference image           |           (b) Translated by 1%           |           (c) Mean image           |

Figure 4: Effect of small translations on metrics. A 1% translation (b) of the reference image (a) appears nearly identical to human observers, yet dramatically affects PSNR and SSIM values. The mean image (c), despite being unrecognizable compared to the reference, produces PSNR and SSIM values remarkably similar to those of the 1% translation.

Perceptual metrics, such as DISTS and LPIPS, demonstrate better robustness to these small translations. DISTS shows exceptional stability with a value of 0.079 for a 1% translation compared to 0.859 for the mean image. This characteristic is especially relevant for evaluating view synthesis, where geometric inaccuracies can manifest as small shifts between synthesized and ground truth views. Since novel view synthesis must address both geometric and appearance errors, metrics that

Table 2: Sensitivity of metrics to small image shifts. Different metrics exhibit varying sensitivity to small spatial misalignments.

| Comparison | DISTS↓ | LPIPS↓ | PSNR↑ | SSIM↑ |
|---|---|---|---|---|
| Translated (0.1%) | 0.008 | 0.059 | 21.3 | 0.623 |
| Translated (1.0%) | 0.079 | 0.491 | 11.2 | 0.375 |
| Translated (5.0%) | 0.121 | 0.723 | 8.1 | 0.249 |
| Mean Image | 0.859 | 0.970 | 10.7 | 0.351 |

can accommodate minor geometric misalignments while still reflecting perceptual quality provide evaluations that correspond more closely to human perception. Based on these findings, we adopted DISTS and LPIPS as our primary evaluation metrics.

## C.2 DEPTH ESTIMATION UNCERTAINTY

Monocular depth estimation is fundamentally ill-posed, as multiple 3D configurations can produce the same 2D image (Poggi et al., 2020). Figure 5 illustrates this ambiguity by comparing depth predictions for an image and its mirror image, a technique similar to that used in left-right consistency for monocular depth training (Godard et al., 2017). The uncertainty map reveals that depth estimators struggle most at object boundaries and in regions with complex geometric structures, such as foliage. When these ambiguous depth estimates are used directly for view synthesis, the resulting images can exhibit visual artifacts as the network attempts to average across multiple plausible depth configurations. Our depth adjustment module, inspired by Conditional Variational Autoencoders (Sohn et al., 2015), addresses this issue by learning a scale map that refines the predicted depth during training, addressing these ambiguities in a way that optimizes for view synthesis quality rather than depth accuracy alone.

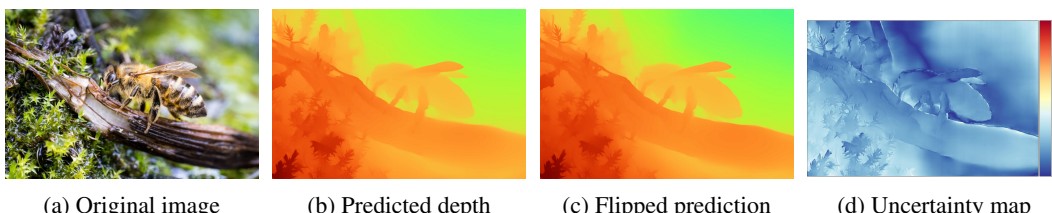

(a) Original image    (b) Predicted depth    (c) Flipped prediction    (d) Uncertainty map

Figure 5: Ambiguity in depth estimation. We demonstrate the inherent ambiguity in monocular depth estimation by (a) taking an original image, (b) predicting its depth using Depth Pro, (c) horizontally flipping the image, applying Depth Pro, and flipping the result back, and (d) computing the relative absolute error between the two predictions to generate an uncertainty map. Higher values (brighter regions) indicate greater inconsistency between predictions.

## D EXPERIMENTS

### D.1 EVALUATION DATASET SETUP

For stereo datasets (Middlebury, Booster), we apply SHARP and the baselines to the left frame and predict the right frame.

For multi-view datasets (ScanNet++, WildRGBD, Tanks and Temples, ETH3D), we proceed as follows:

- For each sequence/scene, we split them into 10-view sets.
- Within each 10-view set, we compute pairwise depth overlap and select pairs with overlap $> 60\%$. For datasets with sparse depth (*e.g.*, ETH3D), we predict monodepth via Depth Pro (Bochkovskii et al., 2025), apply a global scale alignment from dense monodepth to sparse depth (Eq. 16), and compute the monodepth overlap.

- We select $\min(512, \#pairs)$ pairs to evaluate per dataset. For each pair, we predict target image from the source image.

The reason for limiting the number of pairs is the slow inference speed of diffusion-based baselines. For instance, Gen3C takes 15 minutes to synthesize a new view (as a byproduct of synthesizing a video). 512 pairs already take roughly 5 days to evaluate on an A100; any larger set becomes less tractable to evaluate.

Figure 6 shows the distribution of pairwise camera baseline size across datasets.

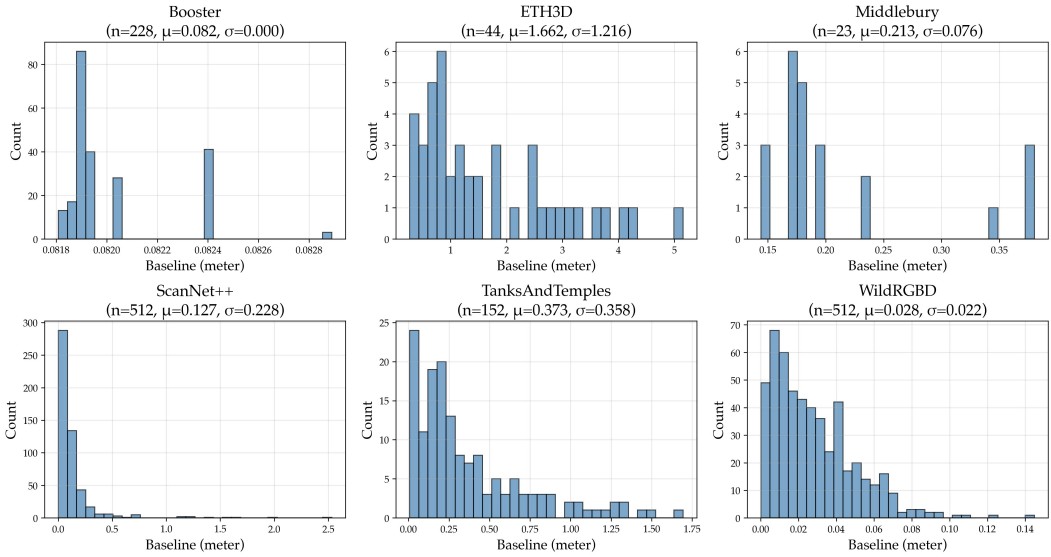

Figure 6: The pairwise camera baseline size distribution across datasets.

**ScanNet++.** We sample from the *nvs test* split with DSLR images.

**WildRGBD.** We sample from the validation split from *nvs list* in each scene.

**Tanks and Temples.** We take the training set for Tanks And Temples. We composite SfM poses and SfM-to-LiDAR transformation, both provided by the authors, to create camera matrices in the metric space, then backproject the LiDAR points to the associated images to form ground truth depth maps. The depth maps were only used for experiments with privileged depth information.

**ETH3D.** We use the ETH3D high-resolution multi-view training set. As with Tanks and Temples, we only use sparse depth for experiments with privileged depth information.

## D.2 BASELINES

We report model sizes of SHARP and baselines in Table 3. The numbers are based on reported numbers in the publication and source code. The original TMPI paper utilizes DPT depth (Ranftl et al., 2021) as the monodepth backbone; we replace it with the latest Depth Pro (Bochkovskii et al., 2025) for better quality and hence report the total parameters with Depth Pro backbone.

Table 3: Parameter counts across models. Trainable parameters are estimated by subtracting frozen module parameter counts from total counts. ∗: finetuning diffusion models.

|  | Flash3D | TMPI | LVSM | SVC | ViewCrafter | Gen3C | SHARP (ours) |
|---|---|---|---|---|---|---|---|
| # total | 399M | 957M | 314M | 2.33B | 3.17B | 7.7B | 702M |
| # trainable | 52M | 6M | 314M | 1.26B∗ | 2.6B∗ | 7.4B∗ | 340M |

To verify that the in-house synthetic data (Section B) is not the dominant factor in the view synthesis fidelity demonstrated by SHARP, we retrain Flash3D on the same in-house synthetic data.

We trained on 24K (3%) and 216K (28%) scenes from our data for 100K steps and 150K steps, respectively. We do not further scale up the number of scenes because we do not find a consistent positive signal of scaling data with Flash3D, and more scenes trigger data loader crashes in the reference implementation[2]. As shown in Table 4, we do not observe a distinct improvement when training Flash3D with our synthetic data. This implies that our in-house data quality is not the principal factor in the reported view synthesis performance.

Table 4: Training Flash3D on in-house synthetic data.

| data | Middlebury | | Booster | | ScanNet++ | | WildRGBD | | Tanks and Temples | | ETH3D | |
|---|---|---|---|---|---|---|---|---|---|---|---|---|
| | DISTS↓ | LPIPS↓ | DISTS↓ | LPIPS↓ | DISTS↓ | LPIPS↓ | DISTS↓ | LPIPS↓ | DISTS↓ | LPIPS↓ | DISTS↓ | LPIPS↓ |
| internal (3%) | 0.325 | 0.599 | 0.335 | 0.403 | 0.398 | 0.630 | 0.235 | 0.417 | 0.453 | 0.756 | 0.506 | 0.673 |
| internal (28%) | 0.433 | 0.647 | 0.442 | 0.415 | 0.488 | 0.696 | 0.255 | 0.448 | 0.553 | 0.815 | 0.570 | 0.686 |
| public (RE10K) | 0.359 | 0.581 | 0.409 | 0.370 | 0.374 | 0.572 | 0.159 | 0.345 | 0.382 | 0.683 | 0.535 | 0.651 |

## D.3 ADDITIONAL QUANTITATIVE EXPERIMENTS

**PSNR and SSIM.** For completeness, we report PSNR and SSIM in Table 5, but we discourage their use for evaluating view synthesis fidelity, as per the analysis in Section C.1.

Table 5: We report PSNR/SSIM metrics for completeness. See Section C.1 for analysis of the metrics.

| | Middlebury | | Booster | | ScanNet++ | | WildRGBD | | Tanks and Temples | | ETH3D | |
|---|---|---|---|---|---|---|---|---|---|---|---|---|
| | PSNR↑ | SSIM↑ | PSNR↑ | SSIM↑ | PSNR↑ | SSIM↑ | PSNR↑ | SSIM↑ | PSNR↑ | SSIM↑ | PSNR↑ | SSIM↑ |
| Flash3D | 15.88 | 0.683 | 22.40 | 0.873 | 18.14 | 0.641 | 18.09 | 0.616 | 15.80 | 0.518 | 15.21 | 0.682 |
| TMPI | 16.42 | 0.688 | 19.44 | 0.833 | 16.16 | 0.712 | 16.44 | 0.559 | 12.41 | 0.368 | 12.61 | 0.540 |
| LVSM | 15.53 | 0.681 | 20.16 | 0.843 | 20.25 | 0.775 | 18.04 | 0.594 | 15.95 | 0.519 | 16.72 | 0.722 |
| SVC | 12.72 | 0.613 | 17.65 | 0.781 | 11.71 | 0.624 | 12.20 | 0.410 | 11.76 | 0.413 | 13.36 | 0.662 |
| ViewCrafter | 10.33 | 0.569 | 14.18 | 0.692 | 13.30 | 0.645 | 14.43 | 0.437 | 11.49 | 0.423 | 11.94 | 0.621 |
| Gen3C | 13.89 | 0.624 | 20.19 | 0.837 | 20.82 | 0.792 | 16.54 | 0.504 | 14.83 | 0.499 | 13.09 | 0.642 |
| SHARP (ours) | 17.12 | 0.693 | 22.19 | 0.864 | 22.63 | 0.833 | 19.57 | 0.655 | 16.33 | 0.528 | 14.51 | 0.610 |

**Runtime.** Runtimes are reported in Table 6. SHARP synthesizes the 3D representation in less than a second on an A100 GPU. The representation can then be rendered in real time (100 FPS or higher on most datasets). We always render the results to the native resolution of the datasets, which explains the variability between datasets (*e.g.* ETH3D has native resolution $6048 \times 4032$).

Table 6: Runtime (in seconds) on an A100 GPU. Note that the SVC/TMPI runtime is lower on ETH3D, since they encountered memory issues and we had to rerun them on an H100.

| | Middlebury | | Booster | | ScanNet++ | | WildRGBD | | Tanks and Temples | | ETH3D | |
|---|---|---|---|---|---|---|---|---|---|---|---|---|
| | Inference↓ | Render↓ | Inference↓ | Render↓ | Inference↓ | Render↓ | Inference↓ | Render↓ | Inference↓ | Render↓ | Inference↓ | Render↓ |
| Flash3D | 0.154 | 0.025 | 0.154 | 0.047 | 0.155 | 0.004 | 0.154 | 0.003 | 0.154 | 0.004 | 0.153 | 0.041 |
| TMPI | 0.328 | 0.249 | 0.333 | 0.248 | 0.315 | 0.247 | 0.183 | 0.294 | 0.272 | 0.218 | 0.222 | 0.157 |
| LVSM | 0.121 | - | 0.120 | - | 0.120 | - | 0.120 | - | 0.120 | - | 0.121 | - |
| SVC | 62.687 | - | 57.598 | - | 62.670 | - | 57.456 | - | 78.846 | - | 32.610 | - |
| ViewCrafter | 119.718 | - | 118.679 | - | 119.385 | - | 119.590 | - | 119.859 | - | 119.922 | - |
| Gen3C | 830.225 | - | 831.775 | - | 836.455 | - | 838.695 | - | 841.418 | - | 838.143 | - |
| SHARP (ours) | 0.912 | 0.010 | 0.911 | 0.016 | 0.911 | 0.006 | 0.912 | 0.004 | 0.912 | 0.005 | 0.910 | 0.022 |

## D.4 EVALUATION WITH PRIVILEGED DEPTH INFORMATION

Table 7 evaluates all view synthesis methods when privileged ground-truth depth maps are used for scale adjustment. We again report PSNR/SSIM metrics for completeness but discourage their use for view synthesis fidelity.

For approaches where a depth proxy is available (Flash3D uses UniDepth (Piccinelli et al., 2024), ViewCrafter uses Dust3r (Wang et al., 2024), TMPI uses DepthPro (Bochkovskii et al., 2025), Gen3C uses MoGe (Wang et al., 2025), and SHARP uses DepthPro (Bochkovskii et al., 2025)), we align the intermediate depth representation $\hat{\mathbf{D}}$ to the ground truth $\mathbf{D}$ to derive an approximate

---

[2]https://github.com/eldar/flash3d/tree/main

global scale factor:

$$s = \text{median}_{p \sim \Omega} \left\{ \frac{\mathbf{D}(p)}{\hat{\mathbf{D}}(p)} \right\}, \tag{15}$$

$$\bar{\mathbf{D}}(p) = s \cdot \hat{\mathbf{D}}(p). \tag{16}$$

For other approaches (LVSM and SVC), for each pair, we apply a linear scale sweep to find the best scale that minimizes the DISTS score.

Table 7: View synthesis fidelity with privileged depth information.

| | Middlebury | | Booster | | ScanNet++ | | WildRGBD | | Tanks and Temples | | ETH3D | |
| | DISTS↓ | LPIPS↓ | DISTS↓ | LPIPS↓ | DISTS↓ | LPIPS↓ | DISTS↓ | LPIPS↓ | DISTS↓ | LPIPS↓ | DISTS↓ | LPIPS↓ |
|---|---|---|---|---|---|---|---|---|---|---|---|---|
| Flash3D | 0.333 | 0.510 | 0.412 | 0.361 | 0.283 | 0.395 | 0.181 | 0.368 | 0.399 | 0.666 | 0.474 | 0.595 |
| TMPI | 0.155 | 0.426 | 0.232 | 0.404 | 0.128 | 0.310 | 0.108 | 0.279 | 0.356 | 0.736 | 0.345 | 0.697 |
| LVSM | 0.243 | 0.564 | 0.294 | 0.428 | 0.125 | 0.236 | 0.088 | 0.229 | 0.219 | 0.558 | 0.456 | 0.668 |
| SVC | 0.181 | 0.518 | 0.257 | 0.381 | 0.146 | 0.459 | 0.120 | 0.407 | 0.199 | 0.653 | 0.410 | 0.700 |
| ViewCrafter | 0.163 | 0.410 | 0.223 | 0.310 | 0.111 | 0.232 | 0.102 | 0.159 | 0.184 | 0.476 | 0.339 | 0.594 |
| Gen3C | 0.124 | 0.347 | 0.192 | 0.291 | 0.085 | 0.196 | 0.078 | 0.118 | 0.149 | 0.434 | 0.283 | 0.568 |
| SHARP (ours) | 0.081 | 0.262 | 0.110 | 0.214 | 0.068 | 0.137 | 0.057 | 0.117 | 0.112 | 0.374 | 0.187 | 0.381 |

| | Middlebury | | Booster | | ScanNet++ | | WildRGBD | | Tanks and Temples | | ETH3D | |
| | PSNR↑ | SSIM↑ | PSNR↑ | SSIM↑ | PSNR↑ | SSIM↑ | PSNR↑ | SSIM↑ | PSNR↑ | SSIM↑ | PSNR↑ | SSIM↑ |
|---|---|---|---|---|---|---|---|---|---|---|---|---|
| Flash3D | 18.61 | 0.719 | 23.16 | 0.879 | 21.98 | 0.803 | 19.45 | 0.679 | 16.40 | 0.567 | 17.06 | 0.674 |
| TMPI | 16.70 | 0.696 | 19.69 | 0.838 | 16.11 | 0.709 | 17.54 | 0.600 | 11.85 | 0.329 | 13.39 | 0.578 |
| LVSM | 15.22 | 0.672 | 19.27 | 0.823 | 23.42 | 0.826 | 19.29 | 0.627 | 16.31 | 0.529 | 16.38 | 0.719 |
| SVC | 15.73 | 0.671 | 20.27 | 0.841 | 15.19 | 0.696 | 14.77 | 0.483 | 13.47 | 0.465 | 13.83 | 0.671 |
| ViewCrafter | 17.11 | 0.703 | 21.55 | 0.860 | 19.73 | 0.788 | 20.10 | 0.672 | 16.84 | 0.566 | 18.81 | 0.721 |
| Gen3C | 18.46 | 0.720 | 23.12 | 0.875 | 22.11 | 0.822 | 22.45 | 0.745 | 17.23 | 0.557 | 18.93 | 0.716 |
| SHARP (ours) | 19.18 | 0.742 | 23.57 | 0.880 | 23.67 | 0.865 | 23.62 | 0.780 | 16.92 | 0.543 | 19.09 | 0.715 |

## D.5 ABLATION STUDIES

We summarize the results from extensive ablation studies in Tables 8–13 and Figures 9–12.

**Datasets.** We report metrics on ScanNet++ (small-scale scenes) and Tanks and Temples (large-scale scenes), and display results on the real-world dataset Unsplash (Unsplash, 2022).

**Models.** For losses, depth adjustment, and unfreezing experiments, we train multiple variants of our model for 60K steps on 32 A100 GPUs only on Stage 1, without Stage 2 SSFT. For the SSFT experiment, we compare Stage 1 and Stage 2 models discussed in Section 4 of the main paper.

**Losses.** We always incorporate color and alpha losses for appearance reconstruction. Our ablation of loss terms (Table 8 and Figure 9) shows that the depth loss reduces geometry distortion, and perceptual loss brings significant improvement in inpainting quality and image sharpness; both losses result in improved metrics. While our regularizers do not move the metrics on the datasets used for ablation analysis, they qualitatively improve scenes with challenging geometry and faraway backgrounds (Figure 9). We also observe that our regularizers boost rendering speed (Table 9), which we attribute to the fact that they prevent degenerate or very large Gaussians.

Because of the importance of the perceptual loss, we separately evaluated the performance improvements from the Gram matrix component (Table 10). Our results show that adding the Gram-matrix loss significantly improves results.

Table 8: Ablation study on loss components. The perceptual loss significantly enhances image quality; regularizer losses ($\mathcal{L}_{\text{reg}} \triangleq \sum_{r \in \mathcal{R}} \lambda_r \mathcal{L}_r$ in Eq. 12) do not have a strong effect on the metrics but yield qualitative improvements. (See Figure 9.)

| $\mathcal{L}_{\text{color}} + \mathcal{L}_{\text{alpha}}$ | $\mathcal{L}_{\text{depth}}$ | $\mathcal{L}_{\text{percep}}$ | $\mathcal{L}_{\text{reg}}$ | ScanNet++ | | | | Tanks and Temples | | | |
| | | | | DISTS↓ | LPIPS↓ | PSNR↑ | SSIM↑ | DISTS↓ | LPIPS↓ | PSNR↑ | SSIM↑ |
|---|---|---|---|---|---|---|---|---|---|---|---|
| ✓ | ✗ | ✗ | ✗ | 0.229 | 0.414 | 18.18 | 0.768 | 0.301 | 0.656 | 14.75 | 0.520 |
| ✓ | ✓ | ✗ | ✗ | 0.162 | 0.270 | 22.95 | 0.844 | 0.239 | 0.548 | 16.23 | 0.550 |
| ✓ | ✓ | ✓ | ✗ | 0.063 | 0.143 | 23.65 | 0.843 | 0.126 | 0.421 | 16.29 | 0.531 |
| ✓ | ✓ | ✓ | ✓ | 0.064 | 0.147 | 22.61 | 0.829 | 0.126 | 0.419 | 16.19 | 0.523 |

**Depth Adjustment.** Table 11 evaluates the contribution of learned depth adjustment during training. The depth adjustment consistently improves perceptual image fidelity metrics. This can also be seen

Table 9: Effect of loss terms on rendering speed. Median rendering latency per frame for different loss combinations. Loss terms improve rendering speed.

| $\mathcal{L}_{\text{color}} + \mathcal{L}_{\text{alpha}}$ | $\mathcal{L}_{\text{depth}}$ | $\mathcal{L}_{\text{percep}}$ | $\mathcal{L}_{\text{reg}}$ | ScanNet++ Latency↓ | Tanks and Temples Latency↓ |
|:---:|:---:|:---:|:---:|:---:|:---:|
| ✓ | ✗ | ✗ | ✗ | 22.2 ms | 15.5 ms |
| ✓ | ✓ | ✗ | ✗ | 12.2 ms | 8.8 ms |
| ✓ | ✓ | ✓ | ✗ | 6.2 ms | 5.6 ms |
| ✓ | ✓ | ✓ | ✓ | 5.5 ms | 4.9 ms |

Table 10: Ablation study on perceptual loss. Adding the Gram matrix loss improves performance.

| Gram loss | ScanNet++ DISTS↓ | LPIPS↓ | PSNR↑ | SSIM↑ | Tanks and Temples DISTS↓ | LPIPS↓ | PSNR↑ | SSIM↑ |
|:---:|:---:|:---:|:---:|:---:|:---:|:---:|:---:|:---:|
| ✗ | 0.070 | 0.153 | 22.26 | 0.827 | 0.130 | 0.441 | 15.89 | 0.517 |
| ✓ | 0.064 | 0.147 | 22.61 | 0.829 | 0.127 | 0.420 | 16.19 | 0.522 |

in the qualitative examples in Figure 10, where the use of the depth adjustment during training yields a model that synthesizes sharper views.

Table 11: Ablation study on depth adjustment. Using the learned depth adjustment module consistently improves image quality. See also Figure 10.

| Learned | ScanNet++ DISTS↓ | LPIPS↓ | PSNR↑ | SSIM↑ | Tanks and Temples DISTS↓ | LPIPS↓ | PSNR↑ | SSIM↑ |
|:---:|:---:|:---:|:---:|:---:|:---:|:---:|:---:|:---:|
| ✗ | 0.077 | 0.154 | 22.89 | 0.838 | 0.148 | 0.444 | 16.04 | 0.519 |
| ✓ | 0.064 | 0.147 | 22.61 | 0.829 | 0.126 | 0.419 | 16.19 | 0.523 |

**Self-supervised Fine-tuning.** Table 12 evaluates the contribution of self-supervised fine-tuning on real images (Stage 2 in Section 3.3). The metrics on the ablation datasets are on par, but qualitative analysis in Figure 11 indicates that self-supervised fine-tuning yields sharper images. We hypothesize that these improvements are due to the limited presence of complex view-dependent effects in synthetic data.

Table 12: Ablation study on self-supervised fine-tuning. While SSFT does not yield consistent metric improvement across datasets, we found it helpful in qualitative studies. (See Figure 11.)

| SSL | ScanNet++ DISTS↓ | LPIPS↓ | PSNR↑ | SSIM↑ | Tanks and Temples DISTS↓ | LPIPS↓ | PSNR↑ | SSIM↑ |
|:---:|:---:|:---:|:---:|:---:|:---:|:---:|:---:|:---:|
| ✗ | 0.063 | 0.142 | 22.86 | 0.835 | 0.125 | 0.433 | 15.91 | 0.513 |
| ✓ | 0.071 | 0.154 | 22.63 | 0.833 | 0.122 | 0.421 | 16.33 | 0.528 |

**Unfreezing Backbone.** Unfreezing the monodepth backbone improves view synthesis fidelity, both quantitatively (Table 13) and qualitatively (Figure 12). Qualitatively, we observe that unfreezing the monodepth backbone resolves boundary artifacts, improves reflections, and resolves artifacts in scenes with challenging geometry.

**Number of Gaussians.** Table 14 evaluates the contribution of the number of Gaussians that we output from our network. We compare the full $2 \times 768 \times 768 \approx 1.2M$ output to a $2\times$ and $4\times$ downsampled output. We see that performance of our method improves when we predict more Gaussians. This is confirmed by our qualitative results in Figure 13.

## D.6 MOTION RANGE

While SHARP excels at generating high-quality nearby views (*e.g.* for AR/VR applications), it was not designed for synthesis of faraway views that have little overlap with the source image.

Table 13: Ablation study on unfreezing the monodepth backbone. See also Figure 12.

| Unfreeze | ScanNet++ | | | | Tanks and Temples | | | |
|---|---|---|---|---|---|---|---|---|
| | DISTS↓ | LPIPS↓ | PSNR↑ | SSIM↑ | DISTS↓ | LPIPS↓ | PSNR↑ | SSIM↑ |
| ✗ | 0.084 | 0.158 | 22.21 | 0.833 | 0.139 | 0.434 | 15.83 | 0.506 |
| ✓ | 0.064 | 0.147 | 22.61 | 0.829 | 0.126 | 0.419 | 16.19 | 0.523 |

Table 14: Ablation study on number of predicted Gaussians. Increasing the number of Gaussians improves performance. See also Figure 13.

| # Gaussians | ScanNet++ | | | | Tanks and Temples | | | |
|---|---|---|---|---|---|---|---|---|
| | DISTS↓ | LPIPS↓ | PSNR↑ | SSIM↑ | DISTS↓ | LPIPS↓ | PSNR↑ | SSIM↑ |
| $2 \times 192 \times 192$ | 0.110 | 0.199 | 20.46 | 0.799 | 0.181 | 0.458 | 16.27 | 0.525 |
| $2 \times 384 \times 384$ | 0.077 | 0.160 | 22.00 | 0.822 | 0.140 | 0.425 | 16.23 | 0.525 |
| $2 \times 768 \times 768$ | 0.064 | 0.147 | 22.61 | 0.829 | 0.126 | 0.419 | 16.19 | 0.523 |

In Figure 7 we study the perceptual metrics trend against the motion values (measured by pairwise camera baseline size in meters) in our evaluation setup (see Section D.1). Experiments show that while SHARP works well, as expected, on small camera motion ($< 0.5$ meters), it retains its quality on larger motion and performs better than most other approaches on extended motion ranges. The SOTA diffusion-based approach Gen3C only outperforms SHARP on ETH3D with motion $> 3$ meters and on ScanNet++ with motion $> 0.5$ meters. We also see that with privileged info, the quality regression over motion range can be further alleviated. In summary, quantitative analysis shows that while the desired motion range is around half a meter, our approach still works reasonably well on larger camera displacement.

In Figure 14 we deliberately extend the range of motion beyond SHARP's intended operating regime. To ensure comparable visual quality of baselines, we provide monodepth from Depth Pro as privileged depth information to all methods in this analysis. Per discussion in Section D.4, we do not show SVC and LVSM results in Figure 14 since they cannot make use of privileged information and cannot perform a scale sweep due to a lack of ground truth novel view.

Qualitatively, we see that extending the range of motion reduces image fidelity in all regression-based approaches. On the other hand, diffusion-based approaches such as Gen3C can synthesize content even for far-away views. However, we also observe the tendency by diffusion models to alter the content of the image even for nearby views (*e.g.*, the stirrups and horse's tail in Figure 14).

We believe it is an interesting research direction to combine the strengths of diffusion-based approaches (synthesis of faraway content) and feedforward models such as SHARP (interactive generation of a 3D representation that can be rendered in real time).

### D.7 FAILURE CASES

Apart from the failure of excessive motion ranges that exceed the operation domain, like all machine learning models, SHARP may fail under challenging scenarios. In Figure 8 we show several such examples.

- In a *macro photo*, due to strong depth-of-field effect, the bee's depth is incorrectly interpreted as behind the flowers, leading to detached wings and distorted tail in novel view synthesis.

- Due to the rich starry texture in a *night photo*, the sky is interpreted as a curvy surface instead of a plain surface far away, causing heavily distorted rendering.

- The *complex reflection* in water is interpreted by the network as a distant mountain; therefore, the water surface appears broken.

These failures are root caused by the depth model, and despite unfreezing the depth backbone, SHARP is unable to recover from the corrupted initialization. We regard this as a long-tail problem of depth prediction. Retraining the depth backbone with higher capacity through more data may

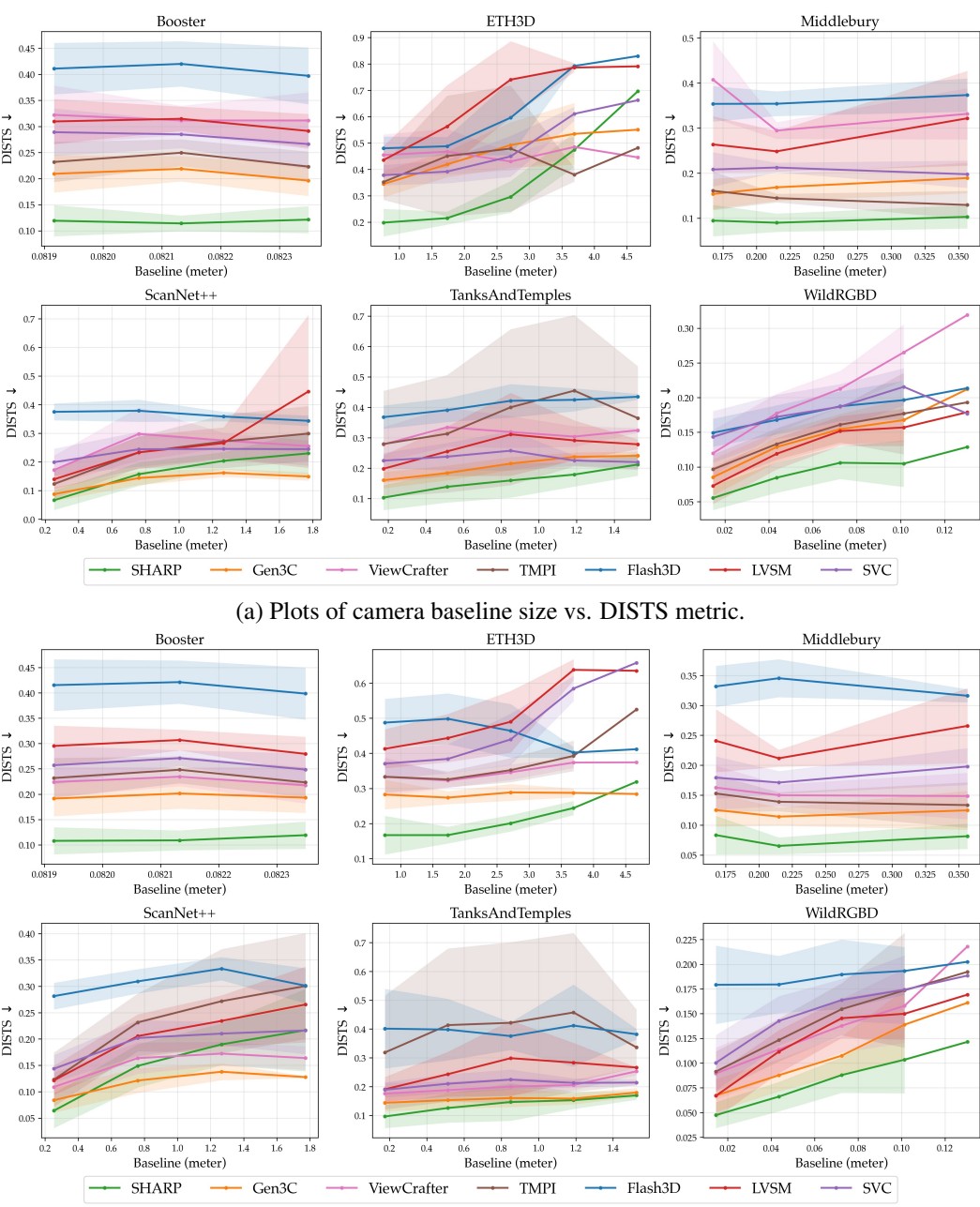

(a) Plots of camera baseline size vs. DISTS metric.

(b) Plots of camera baseline size vs. DISTS metric, with privileged info.

Figure 7: Motion range analysis on the evaluation set. Shade indicates standard deviation. Unshaded data points indicate a single sample in the bin. Bins without samples are skipped, *cf.* Figure 6. SHARP works consistently the best with camera baseline sizes $< 0.5$ meters, and maintains comparable results against diffusion-based approaches on larger motion ranges. It remains the best or the second best up to 3 meters.

alleviate the issue; involving diffusion models with richer priors may be an alternative solution in the future.

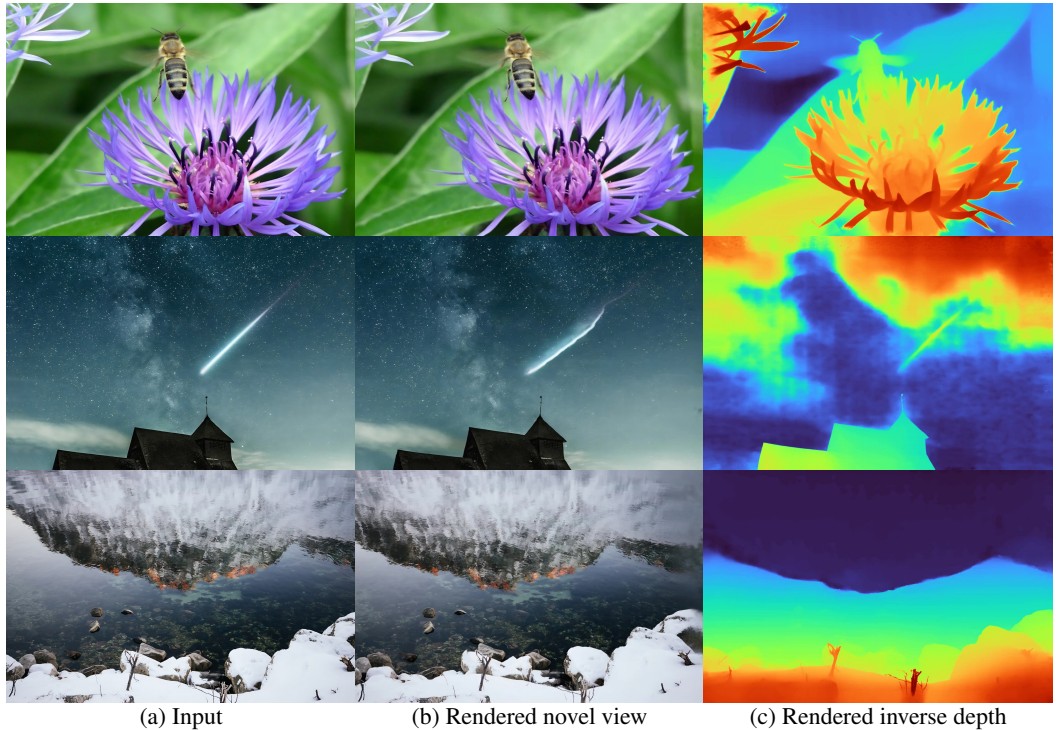

(a) Input      (b) Rendered novel view      (c) Rendered inverse depth

Figure 8: Depth failures in challenging edge cases.

### D.8 ADDITIONAL QUALITATIVE RESULTS

Here we provide extensive qualitative results of all approaches on all datasets in Figures 15–26, both with and without privileged depth information. LVSM, SVC, ViewCrafter, and Gen3C operate at a fixed aspect ratio, therefore we pad their output to match the original image resolution. SHARP consistently produces high-fidelity results. Further video results can be found in `https://apple.github.io/ml-sharp`.

## E LLM USAGE DECLARATION

We used Claude Sonnet 4.5 to polish the writing (e.g. check grammar issues and find better synonyms), to help layout LaTeX tables and figures, and to build the front-end interface of the interactive video comparison.

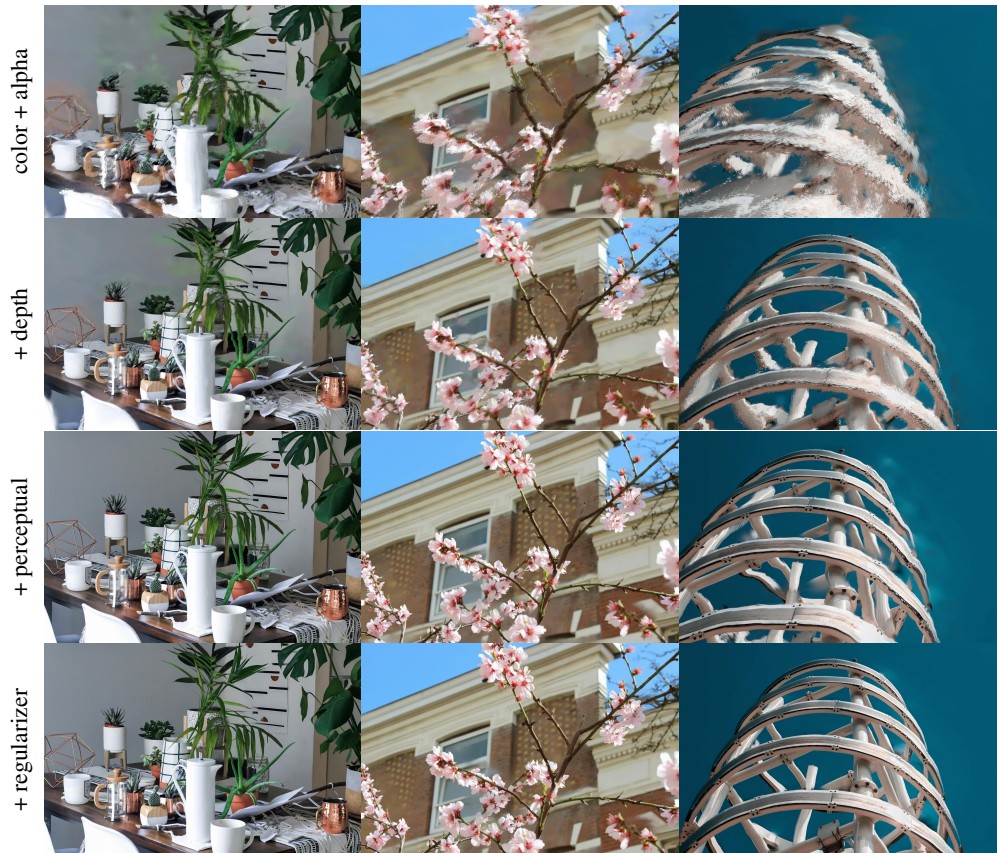

Figure 9: The effect of different loss terms.

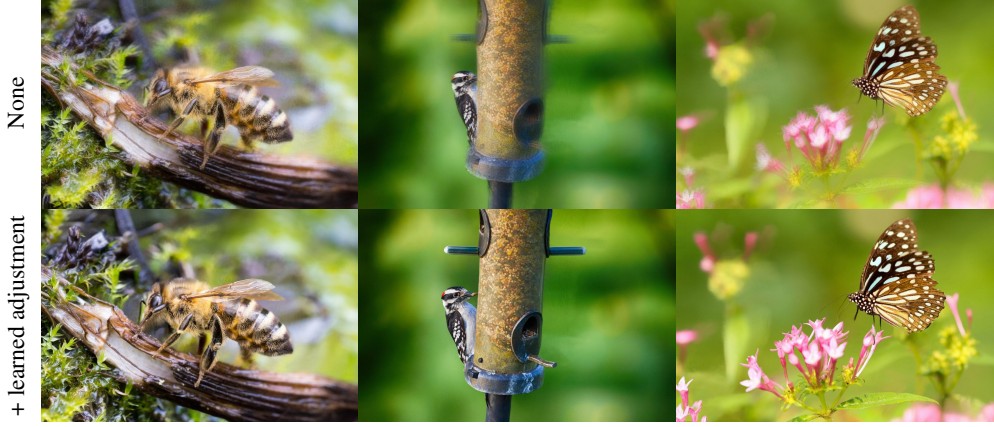

Figure 10: The effect of learned depth adjustment.

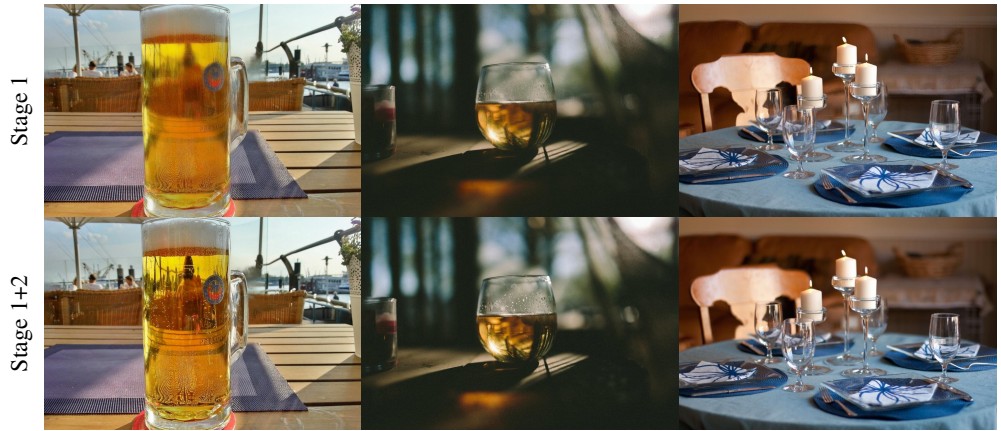

Figure 11: The effect of SSFT.

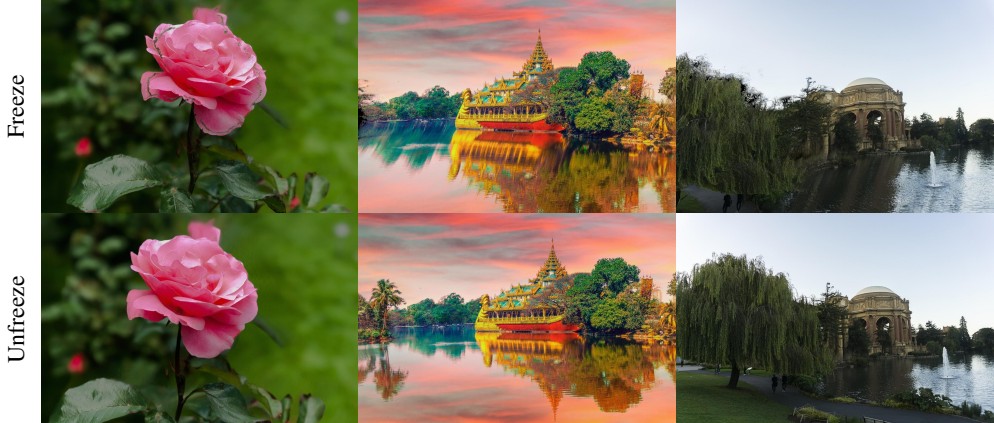

Figure 12: The effect of unfreezing the monodepth backbone.

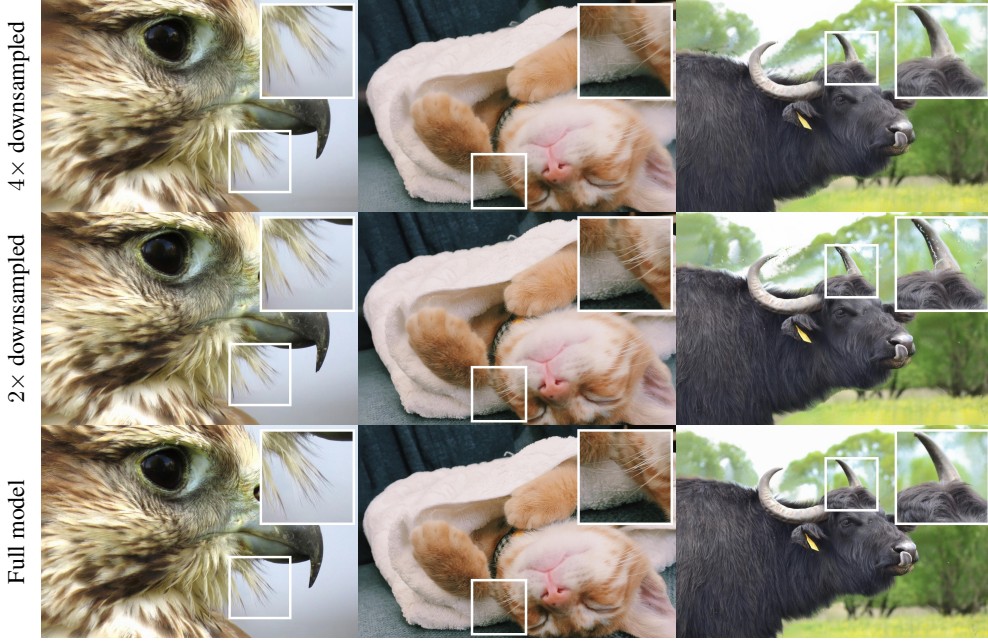

Figure 13: The effect of the number of output Gaussians.

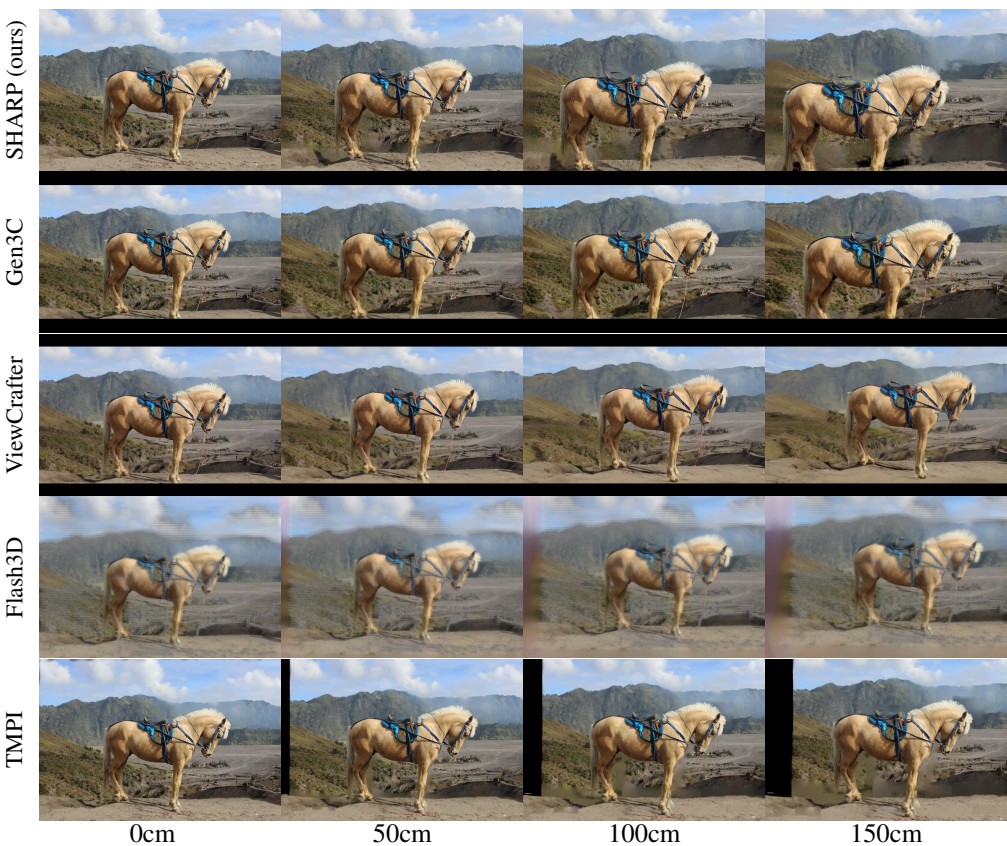

Figure 14: Extending the range of motion beyond nearby views, with monodepth as privileged depth information. We do not show LVSM and SVC as they cannot make use of privileged depth information, see discussions in Section D.4. ViewCrafter and Gen3C operate at a fixed aspect ratio, therefore we pad their output to match the original image resolution.

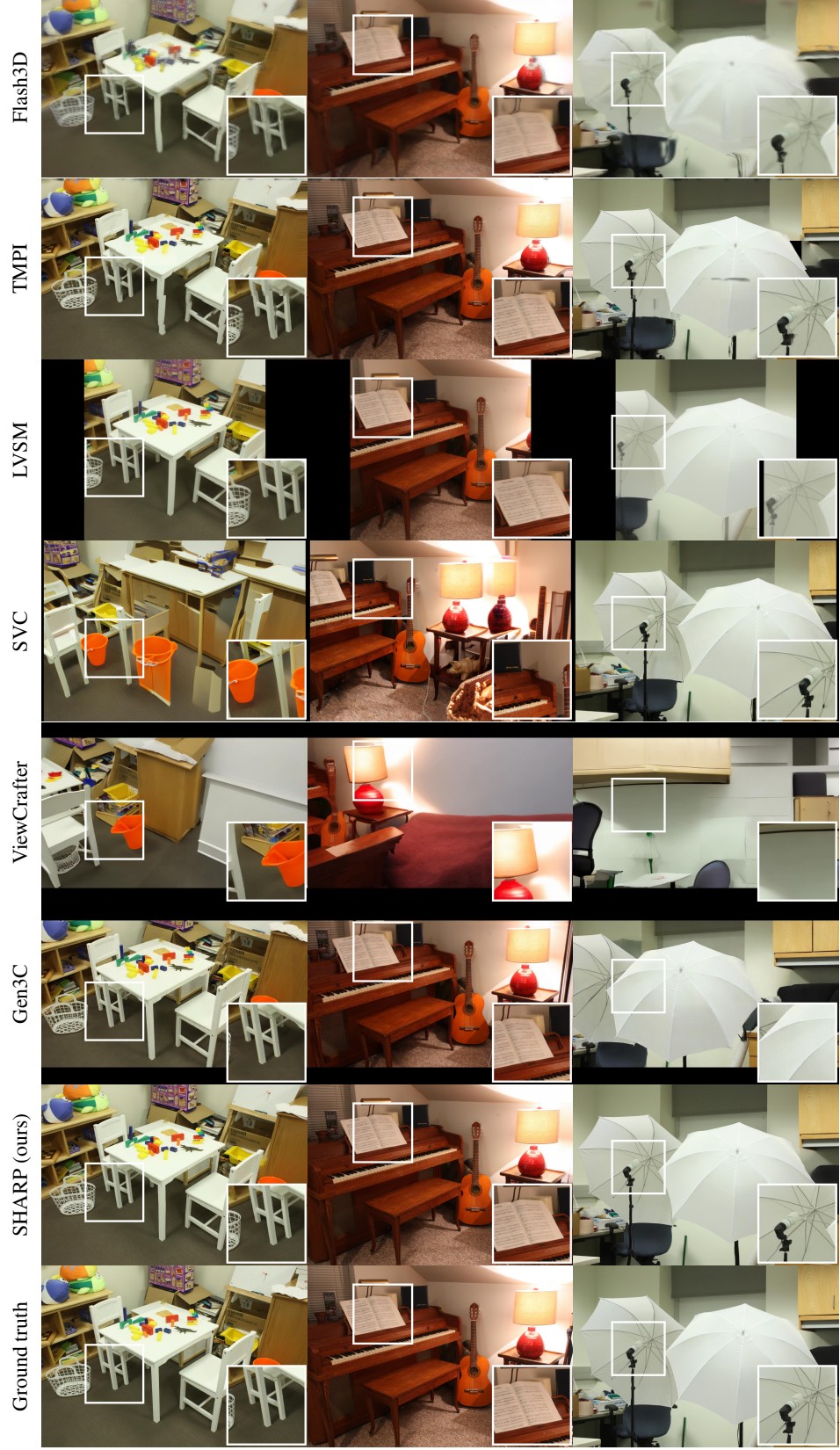

Figure 15: Qualitative comparison on Middlebury.

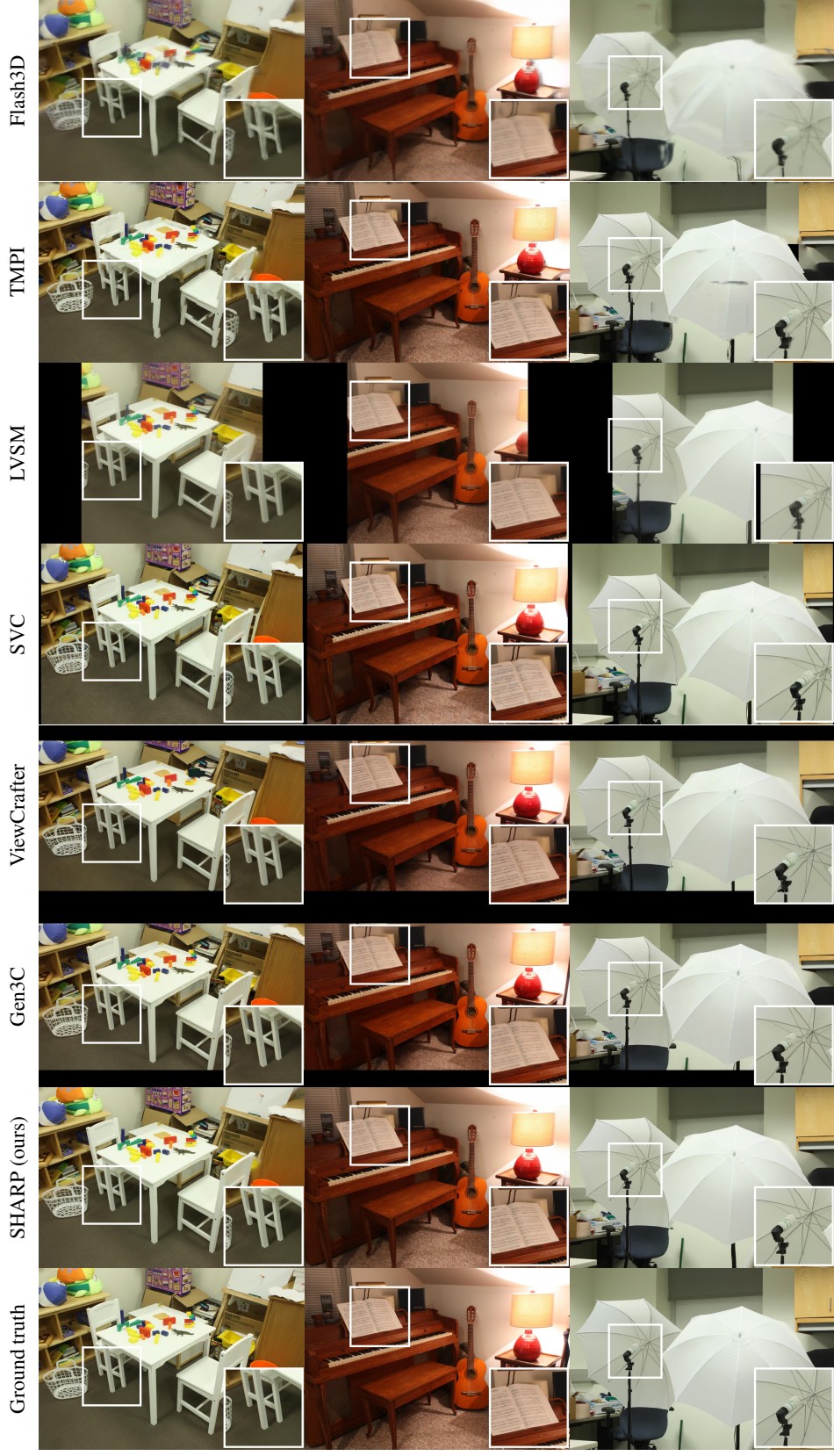

Figure 16: Qualitative comparison on Middlebury with privileged depth information.

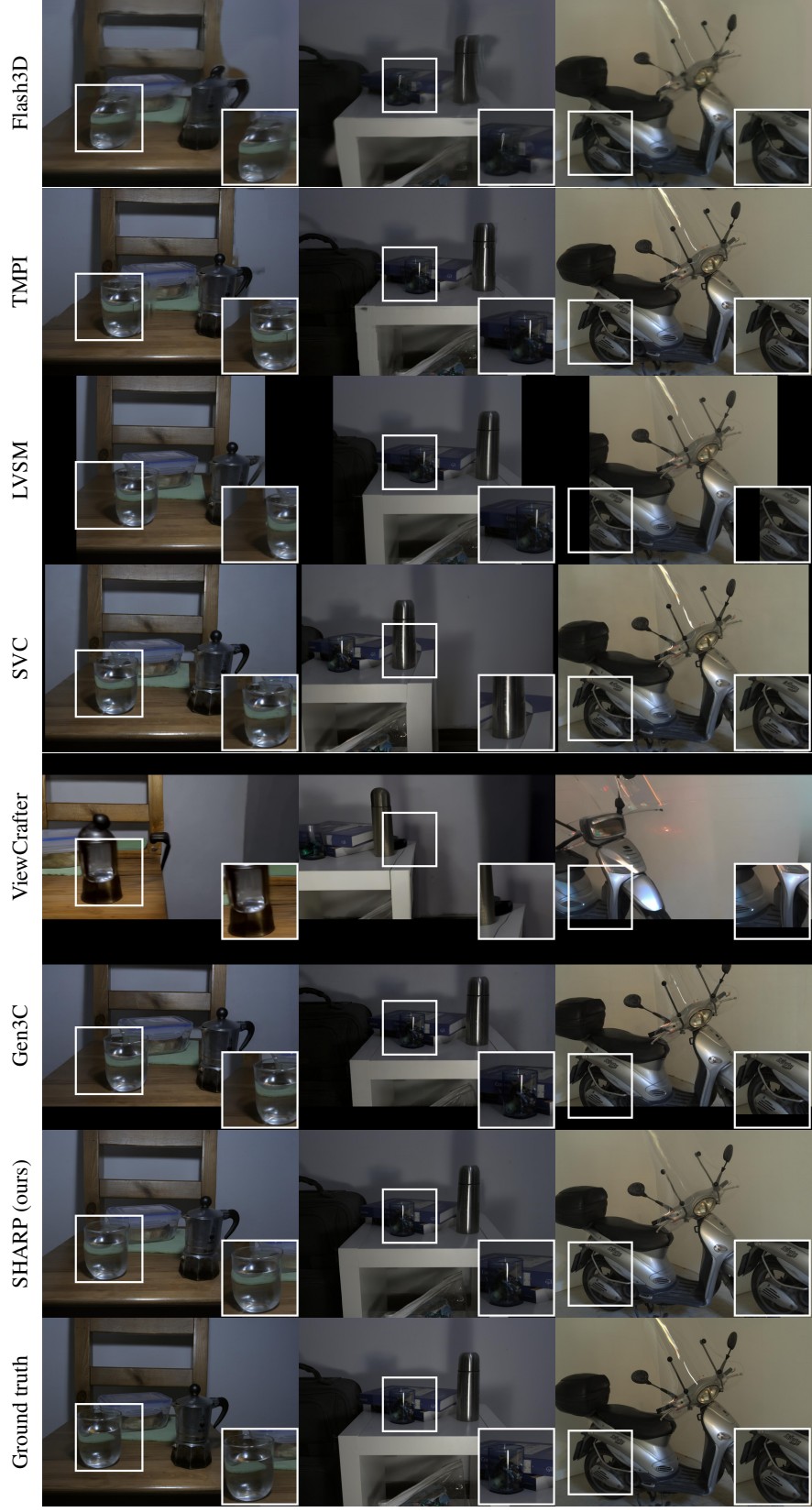

Figure 17: Qualitative comparison on Booster.

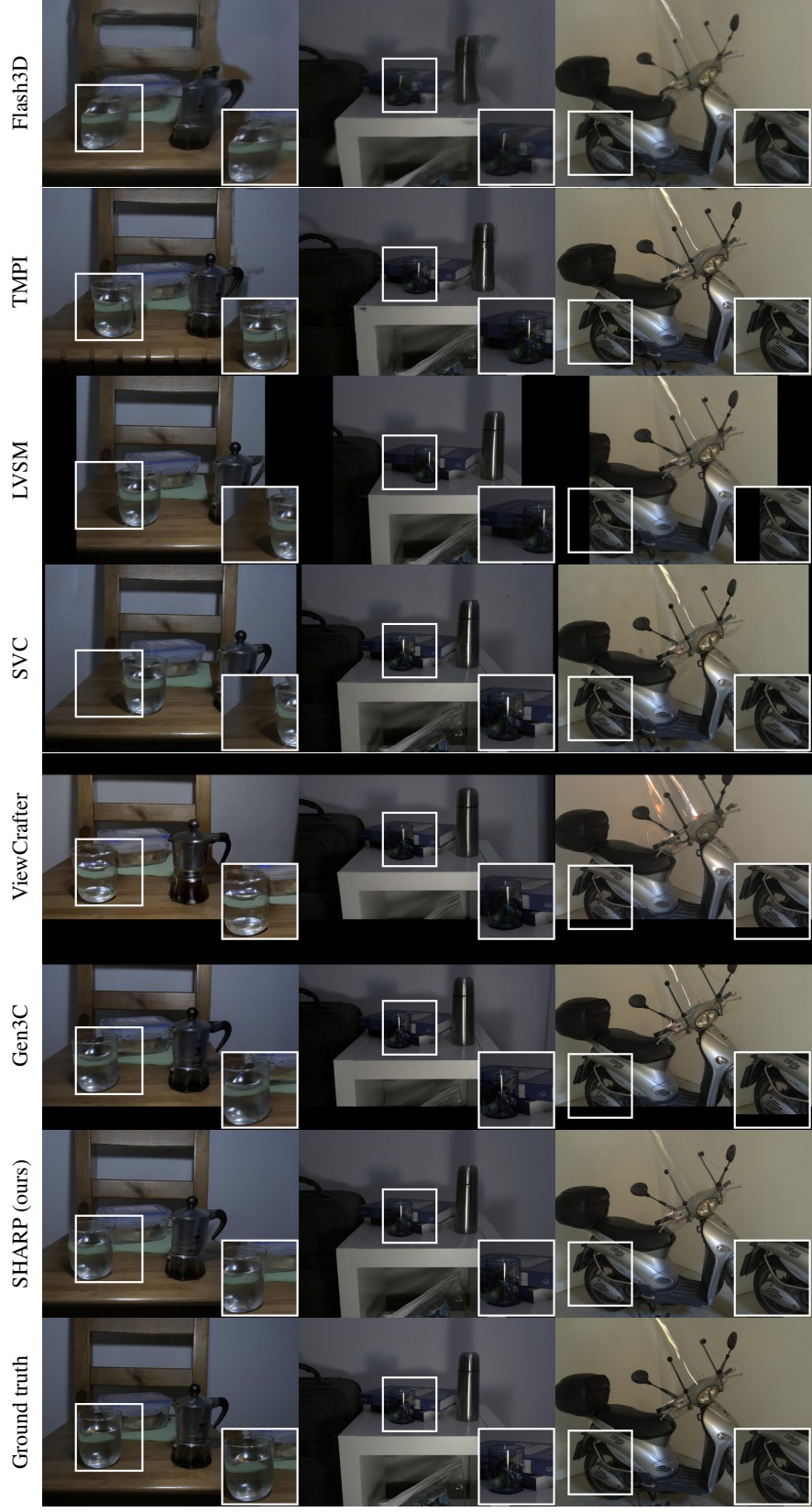

Figure 18: Qualitative comparison on Booster with privileged depth information.

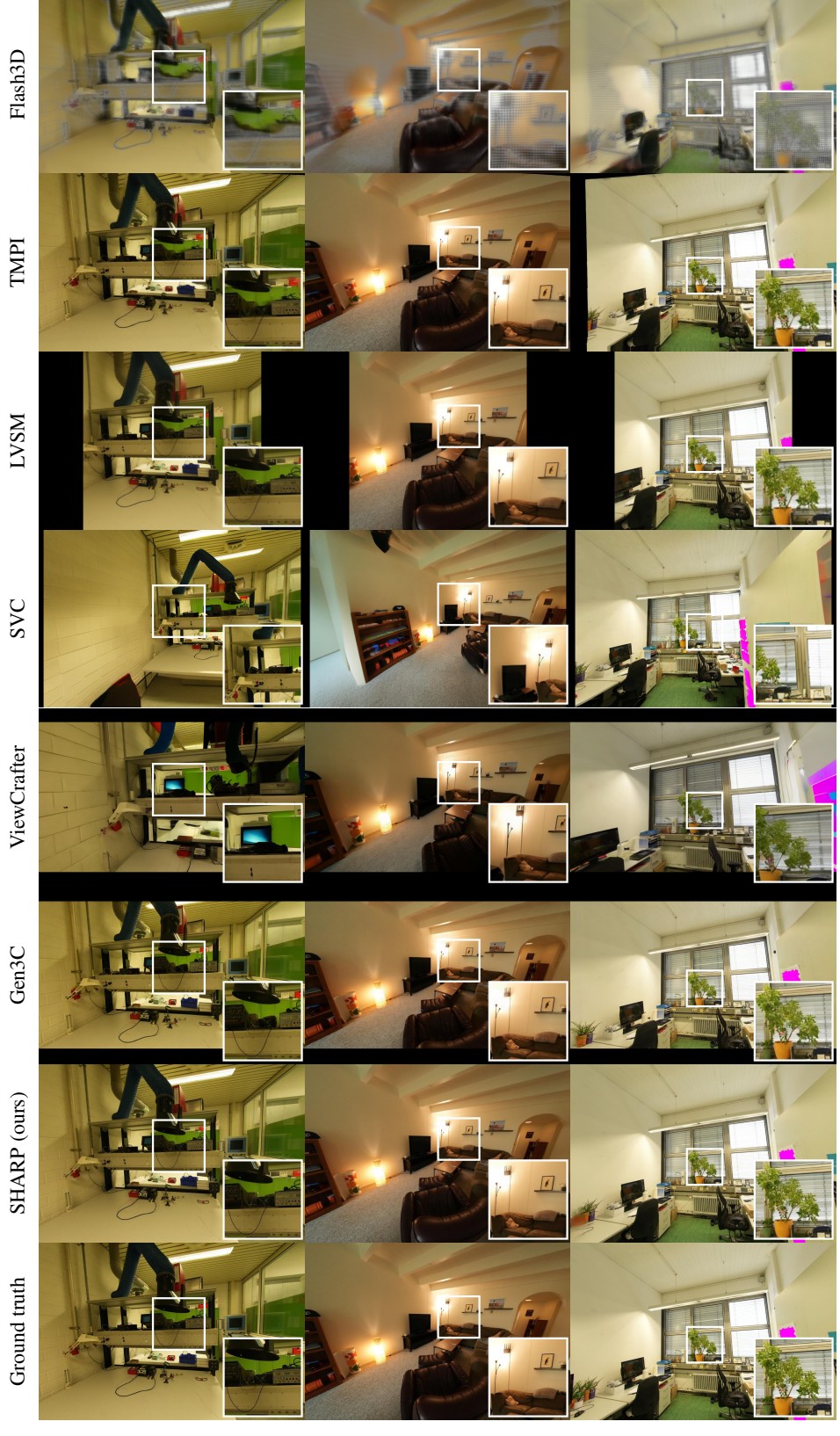

Figure 19: Qualitative comparison on ScanNet++.

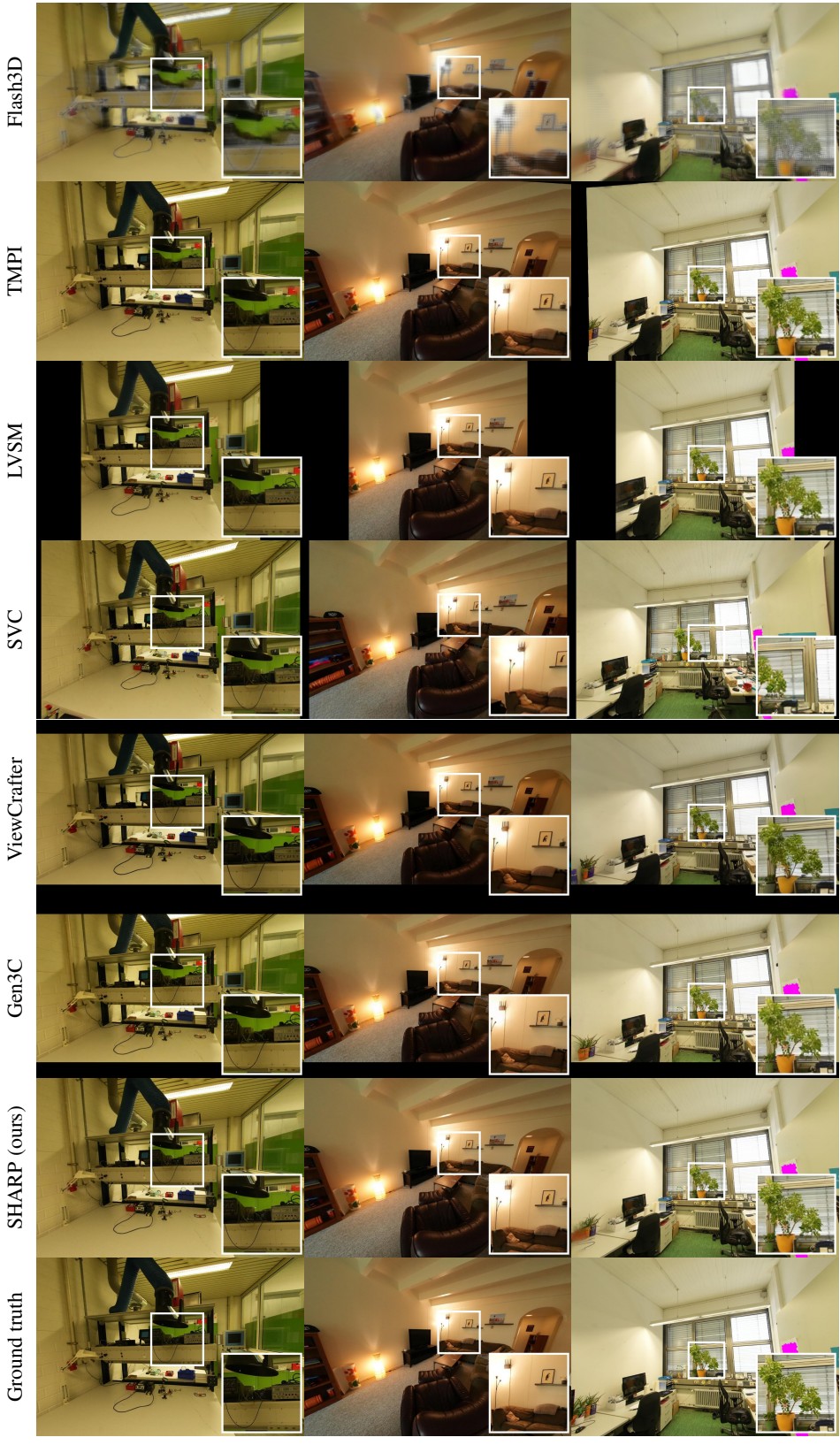

Figure 20: Qualitative comparison on ScanNet++ with privileged depth information.

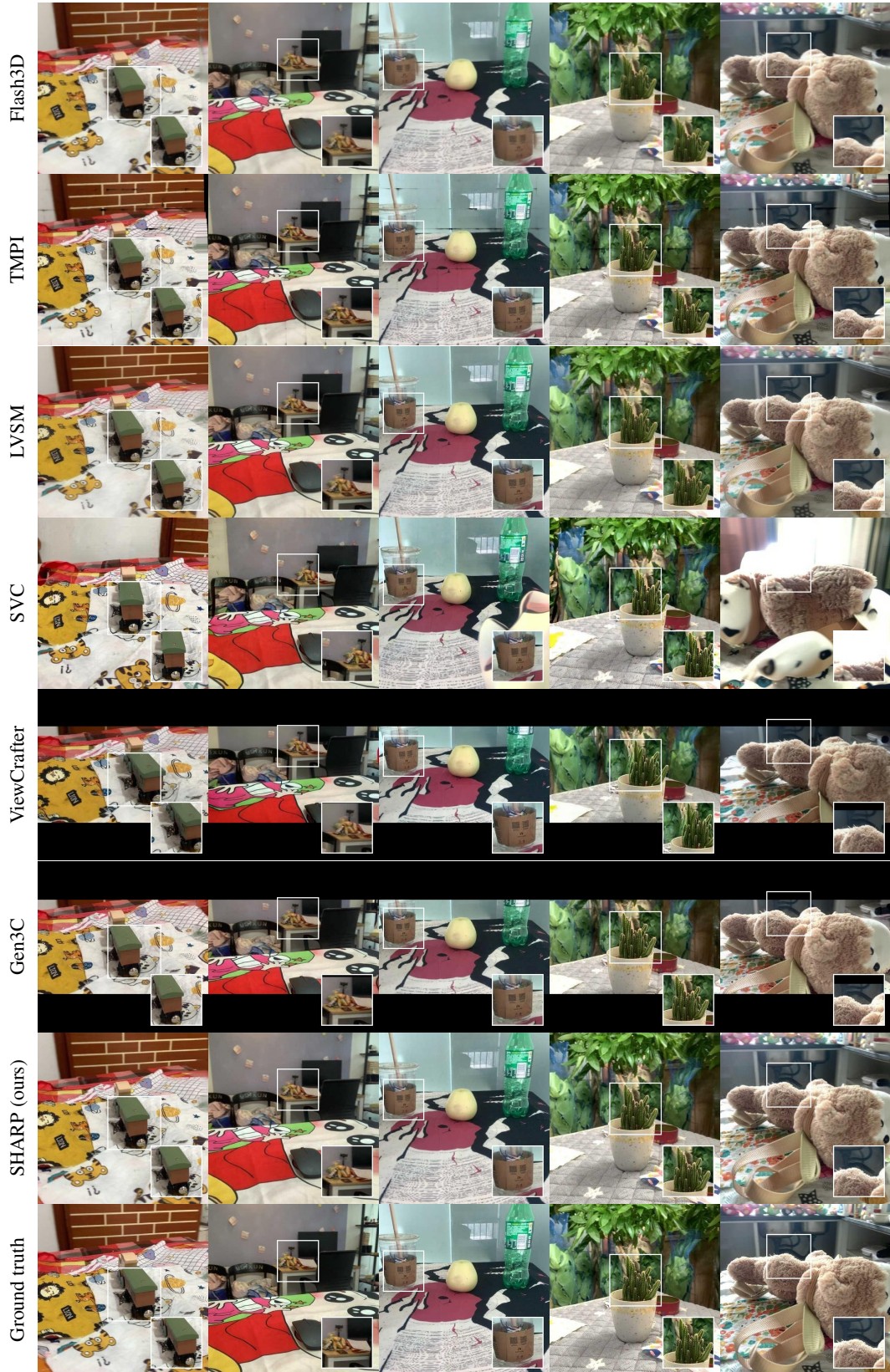

Figure 21: Qualitative comparison on WildRGBD.

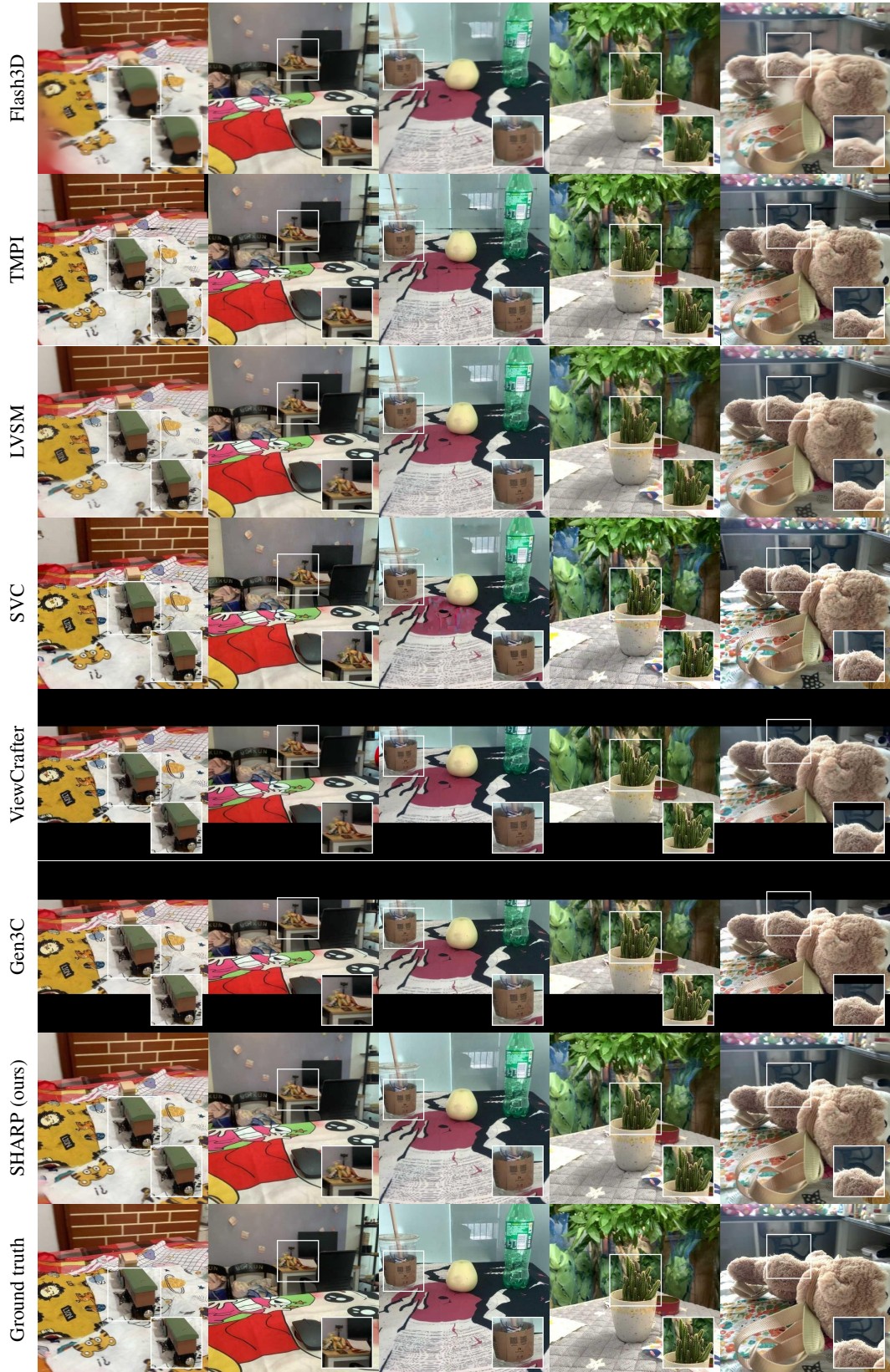

Figure 22: Qualitative comparison on WildRGBD with privileged depth information.

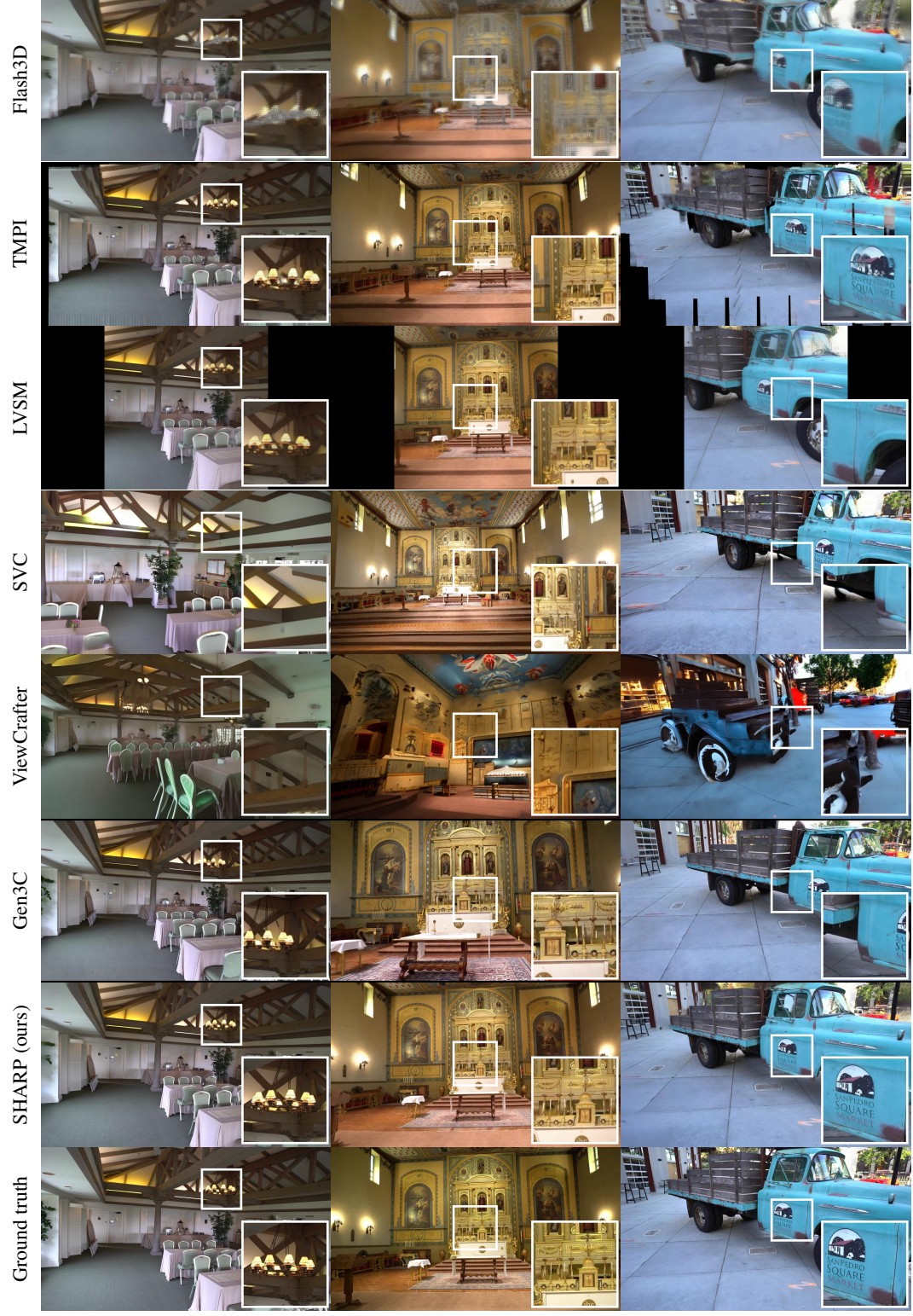

Figure 23: Qualitative comparison on Tanks and Temples.

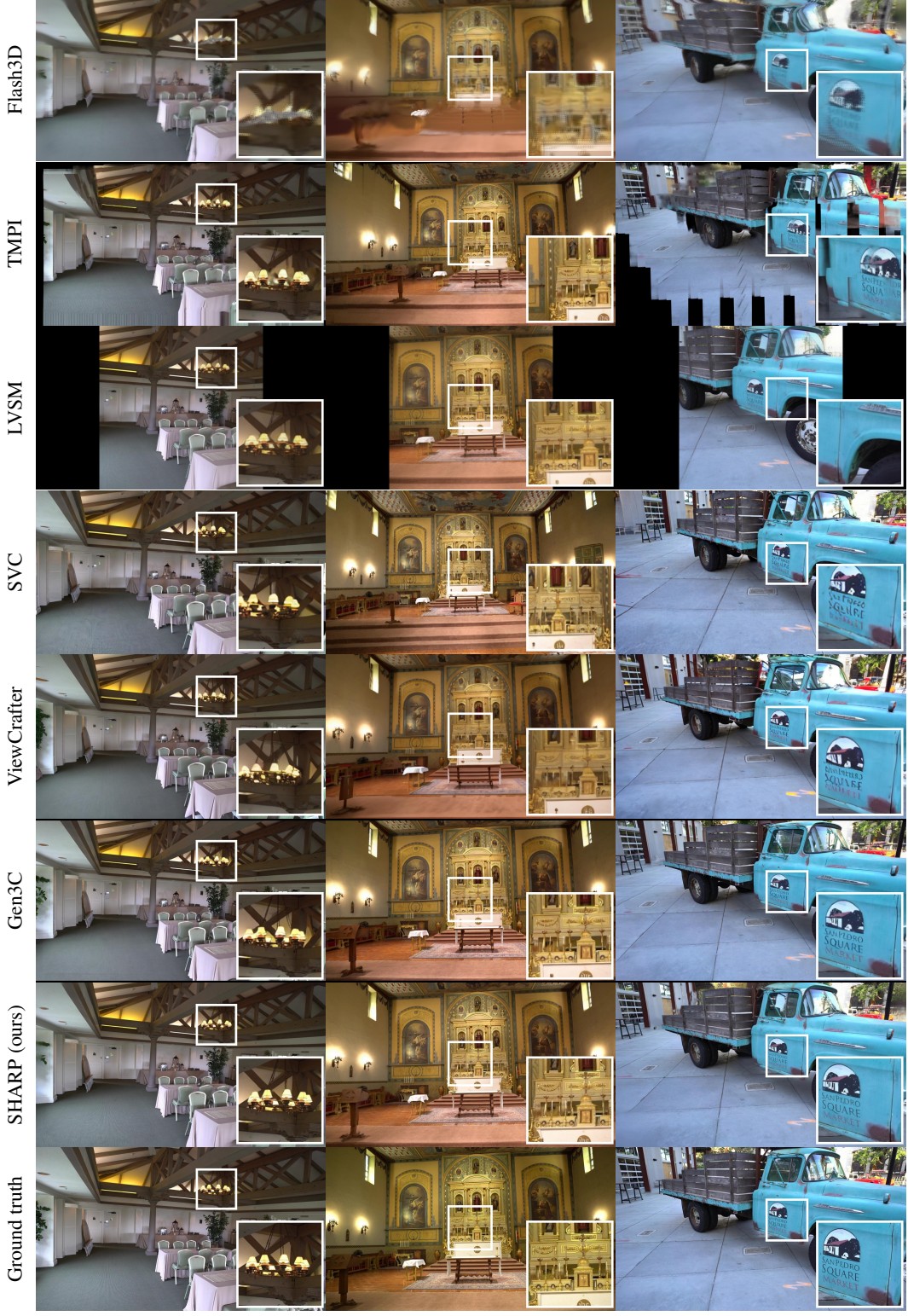

Figure 24: Qualitative comparison on Tanks and Temples with privileged depth information.

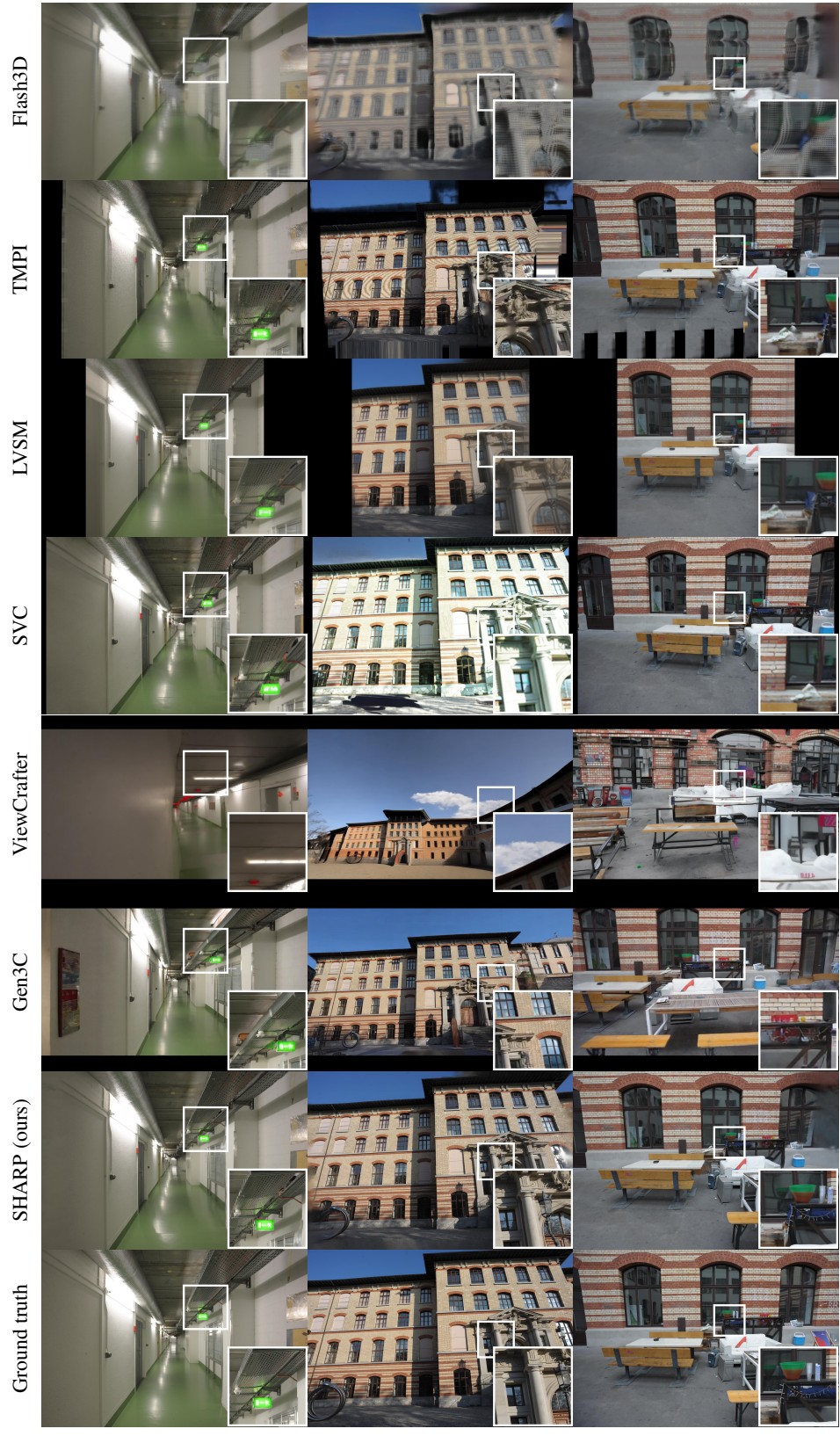

Figure 25: Qualitative comparison on ETH3D.

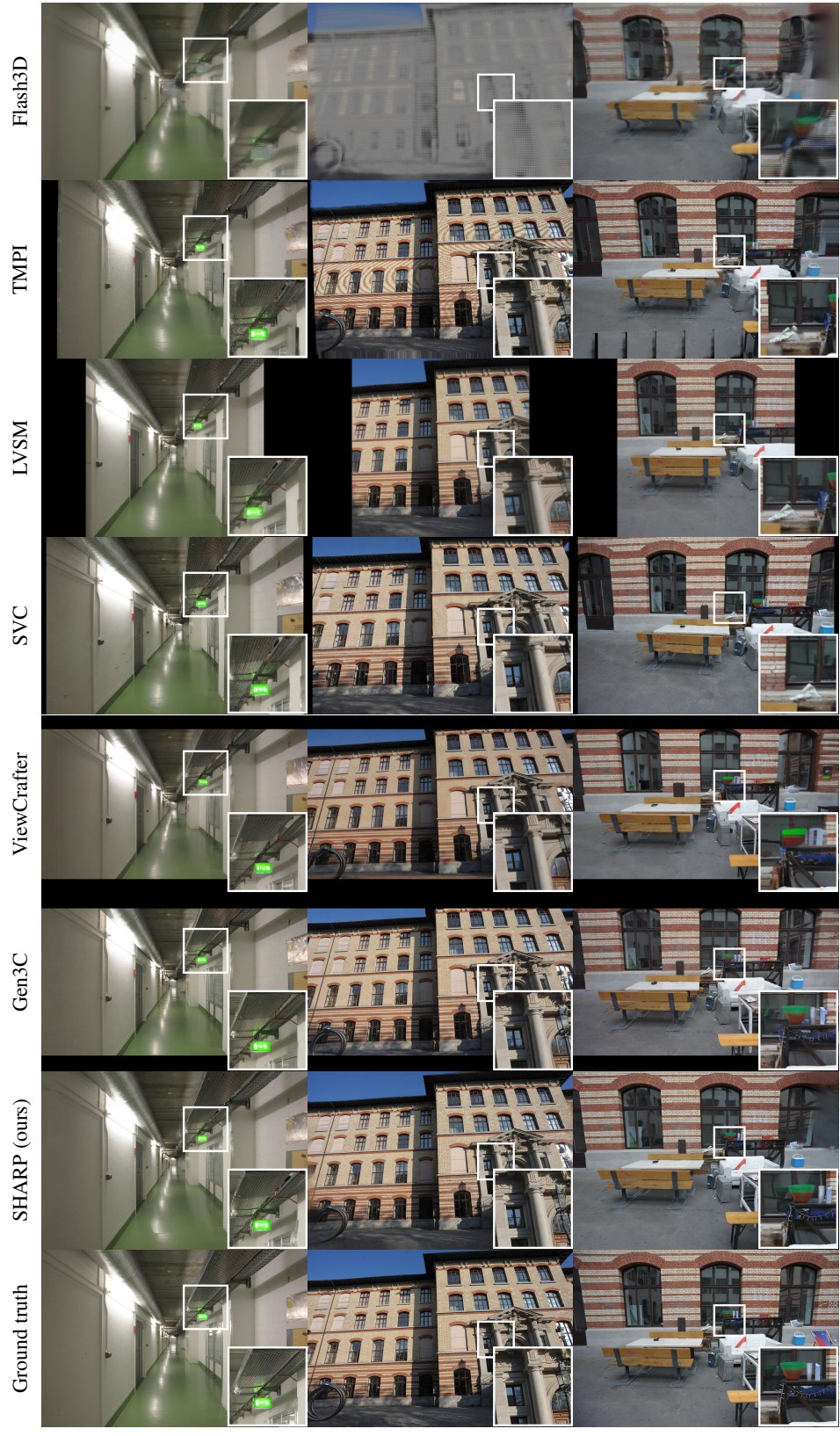

Figure 26: Qualitative comparison on ETH3D with privileged depth information.

