# OpenReview forum: "Sharp Monocular View Synthesis in Less Than a Second"
_ICLR.cc/2026/Conference — ICLR 2026 Poster_

### Official Review · Reviewer_PNr1 · 2025-10-27

**Soundness:** 3
**Presentation:** 3
**Contribution:** 3
**Rating:** 6
**Confidence:** 4

**Summary:**

The paper introduces SHARP (Single-image High-Accuracy Real-time Parallax), a method for photorealistic view synthesis from a single image. SHARP directly regresses a 3D Gaussian representation through a single feedforward pass, achieving real-time high-resolution rendering with metric scale. It combines a Depth Pro encoder, a modified DPT depth decoder, a U-Net-based depth adjustment module, and a Gaussian decoder, trained in two stages—synthetic pretraining and self-supervised finetuning. Experiments on multiple datasets demonstrate strong zero-shot generalization and significant improvements over prior methods in both quality and efficiency. The main contributions include efficient single-image 3D Gaussian regression, depth ambiguity resolution, real-time rendering capability for AR/VR, and establishing a new benchmark for monocular view synthesis.

**Strengths:**

Originality: The primary novel contribution is the formulation of a single, feed-forward network that can directly regress the parameters of a complete, high-resolution 3D Gaussian representation from a single image. This moves beyond prior works that might generate simpler representations or require slow, per-scene optimization. Furthermore, the introduction of a learned depth adjustment module, inspired by Conditional Variational Autoencoders (C-VAEs), is a clever adaptation to handle the inherent ambiguity in monocular depth estimation, optimizing the depth map specifically for the end-goal of high-fidelity view synthesis rather than just metric accuracy. This demonstrates a nuanced understanding of the problem's core challenges.

Quality: The work is technically solid and experimentally comprehensive. Technically, it employs a well-structured architecture that includes a Depth Pro monodepth backbone (with selective unfreezing for task adaptation), a custom Gaussian decoder, and a detailed loss formulation combining color, perceptual (LPIPS), and alpha losses with regularizers for floaters, smoothness, and Gaussian variance to mitigate common 3D reconstruction artifacts. Experimentally, SHARP is evaluated on several datasets (Middlebury, ScanNet++, WildRGBD, etc.) and compared with recent state-of-the-art approaches, including diffusion-based methods. The ablation studies systematically examine key components—such as perceptual loss, depth adjustment, self-supervised fine-tuning, and backbone unfreezing—and the use of perceptual metrics (LPIPS, DISTS) instead of traditional ones (PSNR, SSIM) is appropriately justified.

Clarity: The paper is clearly written and well-structured. The introduction defines the goals—fast synthesis, real-time rendering, and metric scale—while figures effectively illustrate key achievements, such as reduced latency and high-fidelity outputs. The methodology is logically organized, with clear system diagrams and concise mathematical formulations.

Significance: SHARP addresses the challenge of synthesis latency, reducing generation time from minutes to under a second and making real-time single-image view synthesis more practical for interactive applications such as AR/VR. It also sets a strong baseline for feed-forward methods, demonstrating that a carefully designed regression-based approach can achieve high-fidelity nearby views efficiently. These results may guide future research on combining quality and speed in view synthesis.

**Weaknesses:**

-1. In the ablation study for SSFT (Table 11), the quantitative metrics for the model with and without SSFT are very close, with some metrics even slightly worse after fine-tuning.

-2. The model's Stage 1 training relies on a large-scale, in-house synthetic dataset. While the quality of this dataset is likely a key contributor to the model's success, it poses a challenge for reproducibility and fair comparison.

**Questions:**

-1. The paper notes that SHARP is designed for “nearby views.” However, since the term “nearby” can be subjective and application-dependent, could the authors clarify the model’s effective operational range? For instance, the authors could consider plotting key perceptual metrics (e.g., DISTS or LPIPS) against the camera’s baseline distance or angular deviation from the source view, or providing a similar form of analysis to illustrate the model’s effective operational range.

-2. Have the authors encountered cases where the Depth Pro still produces inaccurate estimates even after the depth adjustment module? If so, how does such inaccuracy affect the quality of the generated results, and are the reconstructions still satisfactory under these conditions?

---

> ### Author Response · Authors · 2025-11-21
>
> Thanks for highlighting that our work is **technically solid** and that the **experiments are comprehensive**. We also appreciate the assessment that our paper is **clearly written** and figures effectively illustrate our key achievements. We addressed your comments as follows:
>
> * **Ablation study for SSFT:** We note that SSFT is an optional component of our approach that qualitatively improves results on certain images (Fig. 11 in the supplement). We believe that it is a strength of our approach that it can be trained purely on synthetic data (with optional qualitative improvements from SSFT). We clarified this in the new revision of the manuscript.
> * **Reproducibility**: See *Reproducibility* in general response.
> * **Novel view range**: See *Operating domain* in general response.
> * **Depth inaccuracies:** While our method is very robust, like every machine learning model, we still see failure cases. In challenging scenarios (e.g. macro photo, night photo, and complex reflective surfaces), the encoder of the depth model sometimes fails to generalize, resulting in geometry errors in our final renderings. Larger models and more data may push robustness further. We added those cases in Sec. D.7. and Fig. 8 in the supplement.

---

### Official Review · Reviewer_LvQ8 · 2025-10-30

**Soundness:** 3
**Presentation:** 3
**Contribution:** 2
**Rating:** 4
**Confidence:** 4

**Summary:**

The paper introduces SHARP, a feed-forward method for real-time view synthesis from a single image.
It predicts a 3D Gaussian representation via one forward pass (<1 s on an A100), enabling photorealistic nearby view rendering at over 100 FPS.
SHARP combines a Depth Pro backbone, a learned depth-adjustment module, and a Gaussian decoder trained end-to-end.
Experiments show strong improvements (25–40% LPIPS/DISTS gains) over prior feed-forward and diffusion-based methods.
However the visualization only shows 1-to-1 view synthesis.

**Strengths:**

Novel combination of monocular depth inference and 3D Gaussian Splatting with impressive speed and fidelity.

Clear architecture and training pipeline; loss design and curriculum are well justified.

Extensive comparisons across datasets and perceptual metrics.

**Weaknesses:**

- Novel-view range unclear.

The paper does not specify how far target views are from the input. Report actual displacement (e.g., angle, translation) and analyze performance versus view distance.

- View-to-view consistency.

Since only one novel view for each scene is reported, temporal stability across continuous camera motion is unknown. Evaluating flickering with continuous multiple novel view renderings for frame-to-frame consistency is desired.

- Multi-view generalization.

Can SHARP handle multi-view-to-multi-view synthesis, or is it strictly single-image input due to the monocular depth backbone? Clarify applicability to general NVS pipelines. As most baselines are designed for multiple to multiple NVS, it is a bit unfair setting to directly compare.

- 3DGS necessity.

The model predicts ~1.2 M Gaussians even for small parallax. This appears over-parameterized; justify why a full 3DGS is required. A comparison with pure pixel output like LVSM  under the same training setup would clarify whether 3DGS truly improves quality. And whether 3DGS really ensures 3D consistency under this setting is unclear.

- Reproducibility.

Synthetic data generation and rendering details are not publicly available.

**Questions:**

Please see the weakness.

---

> ### Author Response · Authors · 2025-11-21
>
> Thanks for highlighting the **impressive speed and fidelity** of our approach and praising the **clear architecture** and **well-justified training pipeline**. We appreciate that our **extensive comparisons** across datasets and metrics are convincing. We addressed your feedback as follows in the revised manuscript:
>
> * **Novel view range**: See *Operating domain* in general response.
> * **View consistency:** See *Video comparison* in general response. Our method produces a 3D representation and is hence inherently 3D consistent. This is in contrast to image-diffusion-based approaches such as Gen3C and ViewCrafter or image-regression-based methods such as LVSM. In video comparisons we clearly see that SHARP produces view-consistent outputs while many baseline methods that operate in image space exhibit flickering.
> * **Single view NVS:** SHARP strictly takes one image as input. We believe **single view NVS itself is a legitimate fundamental task for NVS,** akin to the widely accepted monocular depth estimation as a foundation for general geometry perception. It also has real-world applications as discussed in the Introduction. We thank the reviewer for the proposal for studying multi-view generalization of our method. We believe that this feedback shows that our method opens new avenues for the field. We added this idea to the Conclusion section.
> * **Fairness of comparison:** we believe *it may not be accurate to assert that “most baselines are designed for multiple to multiple NVS”.* To the best of our understanding, LVSM is the only compared approach that is not targeted at single view NVS by design:
>     * (i) TMPI and Flash3D are **pure single view NVS** approaches;
>     * (ii) single view NVS is the major operation domain of Gen3C. In the Gen3C paper, the **authors highlighted single view application** as  the **first** component of the teaser and system overview figures, and have a dedicated Sec. 5.2 discussing experimental details on single view NVS;
>     * (iii) Similarly, Stable Virtual Camera **highlighted single view NVS** in their teaser figure, and their project page emphasizes a dedicated single input view gallery after the general gallery.
>     * (iv) ViewCrafter also **highlighted single view Zero-shot NVS as their** **first application** in their project page.
>     * Furthermore, SVC, ViewCrafter, and Gen3C contain 3x, 4x, and 10x parameters respectively compared to SHARP, as reported in Tab. 3 in our supplement. We believe it is fair to compare to **those versatile diffusion-based approaches** that have **much more network capacity**, and **with single view NVS as their core application by design**.
> * **Necessity of 3DGS**: Our method requires a differentiable 3D representation to make it optimizable. A point-cloud representation does not fulfill the requirements. Uploaded video comparisons clearly show the advantage of utilizing 3DGS over pixels: it synthesizes coherent views across viewpoints and produces much sharper details.
> * **Necessity of 1.2M 3DGS**: While 1.2M (2x768x768) Gaussians might be over-parameterized in the multi-view optimization-based setup, we believe they are crucial in single-view feed-forward networks. The network learns to adaptively move the 3DGS around to inpaint holes due to occlusion, adjust fine geometry details for thin structures, and capture as many high-frequency details as possible from the input view. Unlike in multi-view NVS where redundant 3DGS can be pruned with rich supervision from other views, single-view NVS needs this parameterization to faithfully reconstruct the scene.
>     * As a comparison, Flash3D predicts at 0.2M Gaussians (2x256x384), roughly half the resolution of SHARP. From the video comparison we can see a drastic difference in rendering quality, especially when input resolution is high. Checkerboard artifacts are clearly visible with low resolution 3DGS (Flash3D) when the target video is at high resolution.
>     * On the other hand, during inference, 1.2M Gaussians are nothing but a feature map at the resolution of 768x768, and the decoder layers that produce them are efficient Conv/ConvTrans layers, which are not the bottleneck of the network (ViT blocks are). During synthesis, thanks to the efficient 3DGS rasterization, our 1.2M Gaussians can be rendered at 100FPS for most datasets on a standard GPU (supp. Tab. 6). In other words, we are maximizing the full potential of the efficient 3DGS representation by capturing more details, without sacrificing either the inference or rendering speed.
>     * To further underscore the point, we revised Sec. D.5 (Ablation Studies) and added a new ablation study in Tab. 14 and Figure 13 in the supplement that analyzes results when we reduce the number of Gaussians by 4x and 16x.
> * **Reproducibility**: See *Reproducibility* in general response.
> ----
> Edit Dec 03, 2025: update Table IDs to match the final revision.

---

### Official Review · Reviewer_rSXy · 2025-10-30

**Soundness:** 3
**Presentation:** 3
**Contribution:** 2
**Rating:** 4
**Confidence:** 4

**Summary:**

This paper introduces SHARP, a method for photorealistic view synthesis from a single image. It predicts a 3D Gaussian representation of the scene in under one second using a single feedforward neural pass. The resulting representation allows real-time, high-resolution rendering of nearby viewpoints with accurate metric scaling. SHARP achieves state-of-the-art image fidelity, outperforming diffusion-based and feedforward baselines.

**Strengths:**

1. The proposed method is fast and efficient, while achieving high-quality results.

2. The experimental results demonstrate strong performance across multiple datasets and metrics.

3. The writing is clear and the engineering contributions are solid.

**Weaknesses:**

1. The work is more like a system engineering paper rather than a novel research contribution. The scientific novelty is limited. The authors should better highlight the key innovations.

2. It's better that the authors can provide video results to showcase the real-time rendering capabilities.

3. The font used in the paper seems to be non-standard.

**Questions:**

Please refer to the weakness section.

**Details Of Ethics Concerns:**

I encountered two significant issues that I believe may potentially violate ICLR's submission policies, and I am writing to seek your guidance before proceeding with the full scientific review.

The paper appears to deviate substantially from the mandatory ICLR style template. The font, margins, and overall typesetting do not seem to comply with the guidelines.

I carefully checked the submission (including the supplementary materials), and I could not find the required declaration regarding the use of Large Language Models (LLMs) for the paper's preparation. My understanding is that this declaration is a mandatory part of the submission policy this year.

---

> ### Author Response · Authors · 2025-11-21
>
> Thanks for highlighting that our approach is both **fast and efficient** and produces **strong results** across multiple datasets and metrics. We believe that this combination of high-quality results with efficient inference makes our approach particular practical and opens new avenues for future research. To address your comments, we revised the paper as follows:
>
> * **Video comparisons**: See *Video comparison* in general response.
> * **Contributions**: We would like to emphasize the following technical contributions, which we believe are of interest to the broader community: (i) a novel network architecture that can be trained **end-to-end** to optimize novel view fidelity, (ii) a **loss configuration** to enable stable training, and (iii) a novel **depth adjustment module** to address depth ambiguity, a fundamental challenge in view synthesis. In our latest revision, we highlight these points with an itemized list. Our work also demonstrates that high-resolution view synthesis is feasible in a purely regression-based framework.
> * **Style**: See *Style* in general response
> * **LLM usage**: See *LLM usage declaration* in general response.

---

> ### Comment · Reviewer_rSXy · 2025-11-26
>
> I’ve read your rebuttal regarding the contributions (architecture, loss, depth module), but I maintain that this work is primarily a strong piece of system engineering rather than a research breakthrough.
>
>
>
> [Concern 1]: The pipeline integrates existing backbones (Depth Pro/DPT) with 3DGS effectively. While it works well, the design follows a standard regression paradigm and doesn't introduce a fundamentally new mechanism.
>
>
>
> [Concern 2]:Tuning a combination of standard losses is definitely important for performance, but I view this as engineering optimization rather than a methodological novelty.
>
> Depth Adjustment: This is a sensible addition to handle ambiguity, but it feels like a minor tweak within the overall pipeline.
>
>
>
> The paper presents a highly performant system with impressive speed. However, since the core methodology relies heavily on assembling and tuning existing techniques, I find the scientific novelty limited.

---

> ### Author Response · Authors · 2025-11-27
>
> Thank you for your feedback. We appreciate the assessment that “*the paper presents a highly performant system with impressive speed.*” We respectfully disagree, however, with the statement that “the scientific novelty is limited” and its implicit implication on the value of our contributions. The ICLR reviewing guideline explicitly states that “*Submissions bring value to the ICLR community when they convincingly demonstrate new, relevant, impactful knowledge (incl., empirical, theoretical, for practitioners, etc)”.* We believe that our contributions constitute new, relevant, and impactful knowledge in that sense.
>
> We address your specific concerns below:
>
> * **Concern 1:** While our work integrates existing backbones, the key aspect is their effective integration to maximize view synthesis quality while maintaining low latency. The results can be observed in the direct comparison to Flash3D, which is closest to our method. In the quantitative evaluation in Tab. 1, our method reduces DISTS by more than 50% on all datasets and LPIPS by more than 15%. The qualitative results in Fig. 1 show sharper boundaries for our method and ghosting artifacts in Flash3D. This significant performance improvement is the culmination of multiple important design decisions, which we carefully evaluate in Tab. 8-14. One valuable insight is that training the depth backbone alongside the 3DGS decoder lowers the DISTS metric by 15% on average across datasets (Tab. 13). Prior work kept the depth backbone frozen, a technique commonly employed to retain its strong priors and avoid catastrophic forgetting in the presence of a new task, here the prediction of a different modality. We demonstrate that in combination with our other design choices (e.g., depth adjustment, combination of losses), *the backend can be unfrozen to unlock additional quality improvements*. Specifically, we observe that artifacts around boundaries and in scenes with challenging geometry are resolved, and reflections are improved (Fig. 12).
> * **Concern 2:** We respectfully disagree. Prior work focused on simple photometric losses (SSIM, MAE) and did not explore the same combinations as we did. In our work, we not only propose a novel combination, but also carefully analyze how these losses affect results quantitatively (Tab. 8 & 9) and qualitatively (Fig. 9), providing novel insights into maximizing image quality and improving rendering latency. In our view, a novel combination of loss functions that significantly improves image quality (by more than 58%, see Tab. 8) and even lowers rendering latency (by more than 68%, see Tab. 9) constitutes a methodological novelty.
> * **Depth adjustment:** We respectfully disagree. Our controlled experiments in Sec. C.2 demonstrate that the depth adjustment is crucial for achieving sharp results. As we explain in Sec. C.2, “*depth estimators struggle most at object boundaries and in regions with complex geometric structures, such as foliage. When these ambiguous depth estimates are used directly for view synthesis, the resulting images can exhibit visual artifacts as the network attempts to average across multiple plausible depth configurations.*” Optimizing for view synthesis rather than depth accuracy thus requires the proposed depth adjustment module. The results in Tab. 11 confirm its efficacy, and the visual comparison in Fig. 10 shows that it yields sharper images.
>
> In summary, our work demonstrates that existing components can be leveraged for efficient and high-resolution view synthesis and even be boosted to outperform the state of the art through careful integration and optimization. We believe that this is an important empirical insight and hence constitutes a valuable contribution to the ICLR community.
>
> ---
> Edit Dec 03, 2025: update Table IDs to match the final revision.

---

### Official Review · Reviewer_p7mt · 2025-11-03

**Soundness:** 3
**Presentation:** 3
**Contribution:** 3
**Rating:** 6
**Confidence:** 4

**Summary:**

This paper focuses on the task of predicting Gaussian primitives from a given image for novel view synthesis from neighbor viewpoints. The key insight lies in a depth guided framework that predicts Gaussian attribute refinement to a initialized 3DGS. The results outperforms current video-based generative models.

**Strengths:**

1. The task is a promising way for VR/AR applications. This paper focuses on a cutting-edge field.
2. The results are convincing which perform many video based methods that require costly inference.

**Weaknesses:**

1. The authors should provide a video in the supplementary for a more clear comparison with SOTA methods. Since the method outputs a 3DGS, it is more convincing to attach a video showing novel view synthesis results of the 3DGS.
2. How large offset range can the model handle? For the regions that are not visible in the current image, does the model has the capability to generatively infer the occlusions and scene extensions?
3. More recent works like See3D should be compared.

My most concern lies in a lack of direct video showing the quality of GS rendering.

**Questions:**

Please refer to the weaknesses above.

---

> ### Author Response · Authors · 2025-11-21
>
> Thanks for highlighting that our paper produces **convincing results** in a **cutting-edge field**. To address your comments we revised the manuscript as follows:
>
> * **Video comparisons**: See *Video comparison* in general response.
> * **Offset range**: See *Operating domain* in general response.
> * **Occlusions:** Since we trained with a perceptual loss, our method has the capability to produce realistic inpainting in disoccluded regions (Fig. 2). However, as we note in Sec. D6 (Motion Range) of the supplement, our approach is not designed to hallucinate completely new content for faraway views that have little overlap with the source view (Fig. 14 in the supplement). Nonetheless, as described in the conclusion, we believe that our approach paves the way for  exciting new research contributions that marry the strength of our approach (sub-second synthesis of a high-resolution 3D representation) with advantages of diffusion models (realistic inpainting of faraway content).
> * **See3D**: Thanks for pointing out See3D as a potential baseline. We note that we already include Gen3C as a modern diffusion-based baseline, which is concurrent work to See3D (both were submitted to arXiv in March 2025 and presented at CVPR 2025) with similar methodology (compare Fig. 3 of Gen3C and Fig. 4 of See3D). However, for completeness we added See3D to the related work section.

---

### Author Response · Authors · 2025-11-21
**General response to the reviewers and ACs**

We thank all four reviewers for their thoughtful feedback. The reviewers agree that our method produces **strong results** (R1, R4) in a **cutting-edge field** (R1) with **impressive speed and fidelity** (R3) and that our **experiments are comprehensive** (R2, R3, R4). Moreover, it was pointed out that our **writing is clear** (R2, R3, R4) and our **work is technically solid** (R2, R4).
We carefully studied all feedback and incorporated it into the latest revision of the manuscript:

* **Video comparison:** As suggested by R1 and R2, we created an anonymous website to perform interactive video comparisons between our method and the baselines: **[https://anony-monkey.github.io](https://anony-monkey.github.io/)**.
    As can be seen in the comparisons, views generated by our method are much more detailed than those of competing approaches. The videos further qualitatively demonstrate superior consistency across views, addressing feedback from R3.
* **Operating domain:** As suggested by R1, R3 and R4, we revised Sec. D.6 (Motion Range) and incorporated a detailed analysis of the operating domain of our approach into the supplement. We find that SHARP consistently excels in the **0.5 meter** range on all the datasets. Outside this range, SHARP undergoes a gradual degradation in quality, but still remains **the best or second best approach up to a 3 meter baseline**. A qualitative example of the degradation is shown in Fig. 14 in the supplement.
* **Ablations:** As suggested by R3, we revised Sec. D.5 (Ablation Studies) in the supplement and added a new ablation showing that the 1.2M output Gaussians are required for high-fidelity results. We also note that our approach produces these 1.2M Gaussians in less than 1 second, and renders them at 100 FPS on a standard GPU. This shows that the number of Gaussians is reasonable, both achieving real-time rendering and maximizing synthesis quality.
* **Reproducibility**: As suggested by R3 and R4, we note that our method only relies on synthetic and monocular images and there are many open source renderers that can be used to reproduce high quality synthetic datasets (e.g. [Infinigen](https://infinigen.org/)). We also would like to stress that we retrained Flash3D on our internal synthetic data and it does not produce meaningfully better results than the public model weights  (Tab. 4 in the supplement). Our in-house 3DGS renderer is fully compatible to the open source [gsplat](https://github.com/nerfstudio-project/gsplat) framework with comparable rendering speed. We will open source the SHARP model weights and the pipeline compatible to gsplat renderer to further facilitate reproducibility.
* **Style:** As pointed out by R2,  our font and margin did not fully match the official ICLR style. While we used the correct style and template files (.sty and .bst) provided [here](https://github.com/ICLR/Master-Template/tree/master/iclr2026) as per guideline, we found that we did not include [font option](https://github.com/ICLR/Master-Template/blob/a28d335b0d46a3c39b205704a65faf41c9748433/iclr2026/iclr2026_conference.tex#L3) and the [margin patches](https://github.com/ICLR/Master-Template/blob/a28d335b0d46a3c39b205704a65faf41c9748433/iclr2026/iclr2026_conference.tex#L46-L47) in the example tex. In addition, our local compiler overwrote the bundled [fancyhdr](https://github.com/ICLR/Master-Template/blob/master/iclr2026/fancyhdr.sty) package (v3.2) with a more up-to-date local package (v4.1). Those combined resulted in the style discrepancy. After fixing those, our main manuscript length is reduced to 466 lines (prior to revision), well within the 9 page limits, and strictly matches the official style. We incorporated the changes into the revised manuscript.
* **LLM usage declaration**: We only used LLMs for polishing writing (check grammar issues, find better synonyms, lay out LaTeX tables and figures, etc) and for frontend development of the new interactive video comparison website. We added the declaration in Sec. E in the supplement in our revised manuscript.

For more details, please see our individual responses to each reviewer.

---

### Author Response · Authors · 2025-12-03
**Summary of the discussion period**

Following the discussion period, we prepared a final revision of our manuscript, addressing all reviewer feedback.
In the initial submission, reviewers already and consistently appreciated our **strong results** (R1, R4) in a **cutting-edge field** (R1) with **impressive speed and fidelity** (R3), and our **comprehensive experiments** (R2, R3, R4). They had further highlighted the **clarity of writing** (R2, R3, R4) and the **technical soundness** of our work (R2, R4).

The final revision now fully resolves any remaining concerns the reviewers had:

* **Video comparison:** Addressing the main concern of R1, we created an anonymous website to perform interactive video comparisons between our method and the baselines: [https://anony-monkey.github.io](https://anony-monkey.github.io/) The comparisons clearly demonstrate that our method produces more detailed results and better view consistency than baselines.
* **Ablations**: We have added new ablations demonstrating: (i) the necessity of the number of Gaussians for optimal performance (R3), (ii) a detailed analysis of the operating domain (R3), and (iii) a new ablation on the components of our novel perceptual loss, confirming its necessity for best performance (R2).
* **Style**: A major concern by R2 had been on stylistic aspects as our initial submission did not fully align with the ICLR style despite using the provided template, and missed an LLM usage declaration. With our revision both aspects have been fully resolved as our paper now fully aligns with the ICLR style (see previous response for further details) and contains an LLM usage declaration (Sec. E in the supplement).
* **Contributions**: Addressing the main concern of R2, we now better highlight our contributions in the paper, namely: (i) a novel network architecture trainable end-to-end for optimizing novel view fidelity, (ii) a loss configuration enabling stable training, and (iii) a novel depth adjustment module to address depth ambiguity. As demonstrated through careful ablations, all our contributions are non-trivial and essential for optimal performance. Based on these contributions, our model sets a new state of the art on multiple datasets, reducing LPIPS by 25–34% and DISTS by 21–43% versus the best prior model, while lowering the synthesis time by three orders of magnitude.

We are grateful for the reviewer's comments, and believe that addressing all their concerns in the final revision improved our submission considerably. Unfortunately, due to the freeze of reviewer ratings, the reviewers however had no chance to update their ratings to reflect that their concerns have been fully addressed.

---

### Meta-Review · Area_Chair_siYf · 2025-12-23

**Summary:**

The reviewers generally agree that the paper presents a highly efficient and high-quality system for single-image novel view synthesis, achieving state-of-the-art fidelity with extremely fast inference via feed-forward 3D Gaussian regression. Strengths consistently noted include impressive speed, strong quantitative and qualitative results.

The main concerns relate to novelty and positioning. Reviewers view the method primarily as a strong system integration of existing components rather than a clear methodological breakthrough. Additional concerns included missing video evidence, unclear operating range, justification of model size, reproducibility.

**Reviewer Concerns:**

**Addressed by the rebuttal:**

Added video comparisons demonstrating view consistency and quality.

Clarified operating range, robustness, and failure cases.

Added ablations justifying loss design, depth adjustment, backbone unfreezing, and number of Gaussians.

Clarified reproducibility and open-sourcing plans.

**Still outstanding:**

Debate over novelty vs system engineering remains for at least one reviewer.

Dependence on in-house synthetic data limits full reproducibility.

**Reviewer Scores:**

Reviewer p7mt:  Likely unchanged (already positive).

Reviewer rSXy: Likely unchanged (borderline due to novelty concerns).

Reviewer LvQ8: Possibly slightly more positive after added analyses.

Reviewer PNr1: Likely unchanged (already positive).

---

### Decision · Program_Chairs · 2026-01-26

Accept (Poster)